# The role of enhanced velocity shears in rapid ocean cooling during Super Typhoon Nepartak 2016

Yiing Jang Yang [1], Ming-Huei Chang [1], Chia-Ying Hsieh [1], Hung-I Chang[1], Sen Jan[1] & Ching-Ling Wei[1]

Typhoon is a major cause of multiple disasters in coastal regions of East Asia. To advance our understanding of typhoon–ocean interactions and thus to improve the typhoon forecast for the disaster mitigation, two data buoys were deployed in the western North Pacific, which captured Super Typhoon Nepartak (equivalent to Category 5) in July 2016 at distances <20 km from the typhoon's eye center. Here we demonstrate that the unprecedented dataset combined with the modeling results provide new insights into the rapid temperature drop (~1.5 °C in 4 h) and the dramatic strengthening of velocity shear in the mixed layer and below as the driving mechanism for this rapid cooling during the direct influence period of extremely strong winds. The shear instability and associated strong turbulence mixing further deepened the mixed layer to ~120 m. Our buoys also observed that inertial oscillations appeared before the direct wind influence period.

[1] Institute of Oceanography, National Taiwan University, No. 1, Sec. 4, Roosevelt Rd., Taipei 10617, Taiwan. Correspondence and requests for materials should be addressed to Y.J.Y. (email: yjyang67@ntu.edu.tw)

Typhoons (called hurricanes in the Atlantic Ocean) that reach East Asia threaten the island, coastal, and inland populations of nearly 1 billion people. Using in situ observations to advance our knowledge of air–sea exchanges during extremely strong winds and, in turn, improve the accuracy of the numerical typhoon forecasts is of particular importance to providing timely warnings to the public for disaster mitigation and for reducing economic loss that results from false announcements. These false announcements are mostly attributed to forecasting errors in a storm's landfall time/location and typhoon wind strength.

Since typhoons can be intensified by obtaining heat from the ocean via air–sea interactions[1], better knowledge of the hydrographic conditions of the upper ocean is the key to improving typhoon forecasts and thus increases the efficiency of disaster mitigation. The evolving processes in the mixed layer of the upper ocean, atmospheric and oceanic heat exchange, inertial oscillations, and cold wakes in the open ocean are crucial to the development of typhoons and successive typhoons, and therefore, these processes are the central focus of many studies using field observations, satellite remote-sensing, and numerical modeling[2–6]. Previous observations in the stratified coastal ocean and associated studies[7,8] have proven that velocity shear-induced vertical mixing during direct wind influence of tropical cyclones is the primary mechanism for rapid upper ocean cooling while the storm wind is still effective. The rapid cooling subsequently caused a reduction in the intensity of some hurricanes translated across the continental shelf of the Mid-Atlantic Bight[7]. Whether a similar rapid cooling occurs during the typhoons' direct wind influence period (forced period hereafter) in deep water has yet to be examined by in situ observations.

Despite the modeling approach, satellite observations, although powerful, are considerably limited by cloud cover, the spatial and temporal resolutions and are only representative of surface conditions[3,9]. The upper ocean's temperature, salinity, and velocity profiles before, during, and after the passage of a typhoon can only be obtained by in situ measurements. However, the high sea state under the influence of extremely strong winds, the reliability of instrumentation, and the unpredictable nature of typhoons prevent direct shipboard measurements and hinder sampling using limited data buoys and autonomous underwater vehicles (e.g., gliders). Successful sampling of the near sea surface atmosphere and the upper ocean during typhoons relies on the correct selection of locations for the deployment of data buoys and/or gliders prior to the passage of typhoons. A more suitable location can be determined using the statistics of historical typhoon tracks.

Although rare, there are notable direct observations of a few typhoons/hurricanes using anchored data buoys. The Bermuda Testbed Mooring in the western North Atlantic observed Hurricane Fabian (Category 4, August 2003), which caused a prominent surface temperature decrease of >3.5 °C, a deepening of vertical mixing to 130 m, and an upper ocean current of up to 1 m s$^{-1}$ after the passing of Fabian[3]. The Kuroshio Extension Observatory in the northwestern Pacific recorded ocean responses to Typhoon Choi-Wan (Category 1, September 2009) at the closest distance of approximately 40 km from the typhoon's center. The observations showed a decrease in the salinity in the mixed layer as the typhoon approached, a rapid temperature drop and salinity increase, and a vertical displacement of 15–20 m in the upper 500 m due to inertial pumping to the left of Choi-Wan, rather than the right, after the typhoon passed[5]. Moored current meters and surface buoys in the northern South China Sea observed Typhoon Kalmaegi (Category 1, September 2014), which showed strong near-inertial oscillation currents with opposite phases between the mixed layer and the thermocline after the typhoon[10].

Despite these high-quality observations, comprehensive observations of the underlying processes during the impact of a super typhoon on the ocean and air–sea exchange, particularly near the typhoon center, are still scarce. Considering that super typhoons have the potential to cause severe damage, accurate typhoon forecasts are needed for disaster prevention. To further advance our knowledge of the upper ocean responses to typhoons and hopefully improve the numerical forecasts of typhoon tracks and intensity in a cost-effective way, functionally improved surface data buoys were deployed in a region with a high probability of typhoon tracks off the coast of Taiwan (Supplementary Fig. 1a) to capture typhoon and particularly super typhoon data during the peak typhoon season of each year since 2015[11]. Note that the typhoon-heavy region is subject to the western boundary current, i.e., the Kuroshio, mesoscale eddies (Supplementary Fig. 1b), and vigorous internal tides generated over the topographic ridges in the Luzon Strait[12–15] (Supplementary Fig. 1c), thus making the analysis of observational data challenging.

A data set of atmospheric and oceanic variables collected by our two buoys during the passage of Super Typhoon Nepartak in July 2016 provides a unique opportunity to gain an in-depth understanding of the evolution of temperature and current responses to extremely strong typhoon winds in the upper ocean and to validate theories and numerical forecasts of super typhoons. By comparing the parameters of other buoy observations listed in Table 1, our two-buoy observations are the only high-resolution measurements that have captured a super typhoon with the buoy locations situated almost directly on the typhoon track. Additionally, a time-lapse camera mounted on the stainless steel tower of the eastern buoy (NTU2 in Supplementary Fig. 1a) recorded images above the sea surface when the eye of Nepartak approached the buoy, thereby providing an on-site look of the sea surface condition (Supplementary Movie 1). The air–sea interface became indistinguishable during the extreme strong wind influence period, thus considerably hampering air–sea flux estimates.

## Results

**Rapid upper ocean response during the forced period**. Time series data of high-resolution air pressure, wind speed, and tidal signal-subtracted oceanic variables (Methods[16]) that were obtained from the two buoys during the impact of Nepartak are illustrated in Figs. 1 and 2 for NTU1 and NTU2, respectively. The closest distance between the eye of Nepartak and the eastern buoy NTU1 during the observation was 19.2 km at 01:50 7 July, with NTU1 on the left of the typhoon track (marked by 1 in Fig. 1a, b). Approximately 10 h later, the eye center approached the western buoy NTU2 with the buoy located 5.9 km to the right of typhoon track at 12:00 7 July (marked by 2 in Figs. 1a and 2a); this occurred when the air pressure was the lowest (911 hPa) and the wind speed decreased quickly from a peak value of 41 to ~25 m s$^{-1}$ (Fig. 2a). West of 124° E, the satellite sea surface temperature (SST) difference, which was obtained by subtracting the pre-storm SST (5 July) from the post-storm SST (8 July), shows a typhoon induced a cold wake approximately along the right side of the typhoon track. The SST in the cold wake was ~2.5 °C cooler than the surrounding water (Fig. 1a). In contrast, east of 124° E, a cold wake was not confined to the right of the typhoon track; this was presumably due to a pre-existing cyclonic eddy[17] characterized by −0.1 and −0.2 m contours of an SSH anomaly (white curves in Fig. 1a; Supplementary Fig. 1b). We focus the discussion on the upper ocean temperature and velocity variation during the forced period, which is defined as the residence duration of the direct wind influence of Nepartak. Note that the strength of Nepartak remained at Category 5 between 06:00 6 July and 12:00

**Table 1 Comparison of deep water buoys observed tropical cyclones**

| Tropical cyclones | | | Mooring characteristics | | | $L_{min}$ | $W_{max}$ |
|---|---|---|---|---|---|---|---|
| Year | Name | C | Location of buoy | Depth | Parameters | | |
| 1975 | Eloise[35] | 2 | Gulf of Mexico | 2541 | $T$ at 2, 50, 200, and 500 m; $(u, v)$ at 50 m. | 16 | 35 |
| 1988 | Nelson[36] | 2 | Southeast of Japan | 4900 | $T$ at 8 depths in the upper 100 m. | 50 | 43.5 |
| 1995 | Felix[3] | 1 | Bermuda | 4567 | $T$ at 7 depths (upper 150 m); $(u, v)$ at 25, 45, 71, and 106 m. | 90 | 38 |
| 2003 | Fabian[3] | 3 | Bermuda | 4567 | $T$ at 16 depths (upper 1500 m); $(u, v)$ profiles (upper 200 m). | 102 | 54 |
| 2005 | Nate[3] | 1 | Bermuda | 4567 | Same as Fabian in 2003. | 123 | 39 |
| 2009 | Choi-Wan[5] | 1 | Kuroshio Extension | 6000 | $T$ at 20 depths (upper 525 m). | 40 | 30 |
| 2010 | Fanapi[9] | 3 | Western North Pacific | 5450 | $T$ at 14 depths (upper 147 m). | 68 | 18 |
| 2010 | Megi[9] | 5 | Western North Pacific | 5500 | $T$ at 10 depths (upper 148 m). | 123 | 23.5 |
| 2014 | Kalmaegi[10] | 1 | South China Sea | 3990 | $T$ at 15 depths (upper 400 m); $(u, v)$ profiles (upper 245 m). | 32 | 23 |
| 2016 | Nepartak | 5 | Western North Pacific | 5490 | $T$ at 12 depths (upper 500 m); $(u, v)$ at 25 and 75 m. | 19 | 46.5 |
| 2016 | Nepartak | 5 | Western North Pacific | 4870 | $T$ at 12 depths (upper 500 m); $(u, v)$ at 25 and 75 m. | 6 | 44 |

Column C means category based on the Saffir–Simpson scale, Depth is the water depth (in m) at where the buoy was anchored, $L_{min}$ is the closest distance (in km) from the buoy to the typhoon/hurricane eye center, and $W_{max}$ is buoy observed maximum wind speed (in m s$^{-1}$)

8 July before it touched the southeast coast of China (Supplementary Fig. 1a). Nepartak moved at a speed of ~14 km h$^{-1}$ (~3.89 m s$^{-1}$) and remained at a constant storm size as it passed by NTU1 and NTU2. The storm's radius during wind speeds of Beaufort scale 7 and 10 were approximately 200 and 80 km (black dashed circle in Fig. 1a), respectively.

We determined that the forced period began as the wind speed increased rapidly to 17 m s$^{-1}$ (marked by the magenta dashed line in Figs. 1b and 2a) and ended as the wind speed decreased to 17 m s$^{-1}$. Our observations find that the evolution of the upper ocean responses before and after the arrival of the typhoon center at the buoy differs significantly and, therefore, the forced period is split into Stages 1 and 2 by the time that the minimum air pressure is recorded by the buoy. The forced period at NTU2 (14.3 h) is longer than that at NTU1 (9.25 h) because NTU2 experienced almost the full diameter of Nepartak's storm circle, whereas NTU1 was on the secant line of the storm circle.

Before the forced period of NTU1, the temperature profile presents a wave-like variation as seen by the 27.2 °C isotherm (thick black curve in Fig. 1c). The background temperature profile (black dashed curve in Fig. 1e) obtained by taking time average of temperature profiles from 13:00 to 15:00 on 7 July shows a thin mixed layer of 7 m (Methods) above a thick thermocline between 7 and 105 m. A second thermocline is seen below 105 m. The wave-like temperature variation showed ~1 °C cooler at time 1 and 0.5 °C warmer at time 2 in the thermocline (denoted by 1 and 2 in Fig. 1c, e), whereas the mixed layer remained constant. At the beginning of Stage 1, temperatures in the mixed layer and thermocline decreased, as shown at time 3 in Fig. 1e. A dramatic change in temperature occurred near the end of Stage 1 at 1:30 as shown by the 27.2 °C isotherm; this is when the mixed layer deepened to 47 m with an ~2 °C temperature decrease, and the thermocline was compressed to ~70 m thick between 47 and 117 m at time 4. The deepening of the mixed layer was accompanied by a cooling in the upper 27 m and a warming below 27 m, which is indicative of turbulent mixing[7,10]. In contrast to previous studies that indicate the heat pump as being a result of subsurface warming that may persist for a long time and plays an important role in the global heat circulation[18], our observations show that the deepening of the mixed layer lasted for only ~3 h in Stage 2. Thereafter, the temperature structure presented only near-surface cooling without subsurface warming as shown at times 5 and 6 in Fig. 1e.

The current meter attached with the mooring line of NTU1 at 75 m observed that, during Stage 1, the direction of the meridional velocity ($V_{curr}$) was opposite to that of the meridional wind ($V_{wind}$), and their coherence was likely weak (Fig. 1d). After Stage 2, the wind direction changed quickly from northerly to southerly and the northward current strengthened from 0.5 to 1.4 m s$^{-1}$. The mixed layer depth deepened to 52 m and the temperature variation extended to 120 m (the bottom of subsurface warming; time 4 in Fig. 1e) presumably due to turbulent mixing.

At NTU2, the temperature profile averaged from 00:00 to 02:00 (black dashed curves in Fig. 2d) and shows a thicker mixed layer of 27 m and a moderate warm anomaly within the thermocline before the forced period (time 1 and 2 in Fig. 2b). At the beginning of Stage 1, the 27.2 °C isotherm deepened (time 3 in Fig. 2b), but the mixed layer depth remained at 27 m. The corresponding cooling and warming occurred concurrently above and below this depth, respectively, near the end of Stage 1 at approximately 11:00. The mixed layer deepened to 67 m with a temperature decrease of ~2 °C at time 4. The observed cooling in the upper 42 m and warming below 42 m is hypothetically attributed to strong wind-induced turbulent mixing. The bottom of the warming reached 142 m at time 4 (Fig. 2d), which is deeper than that observed at NTU1 (117 m). As time progressed to time 5 (Fig. 2b), the mixed layer depth rebounded to 27 m, but the range of the concurrent cooling and warming was confined to the upper 87 m and eventually disappeared at time 6 (Fig. 2d). Notably, temperature was decreased in the upper 300 m at time 6 at both NTU1 and NTU2.

Two current meters were attached to the mooring line of NTU2 at 25 and 75 m; these were used to estimate the shear instability and the source of turbulence production[19,20] in the upper 75 m. Figure 2c shows that $V_{curr}$ at 25 m (thick black curve) varied consistently with the change of $V_{wind}$ during 00:00–11:00 7 July, and the weakening of $V_{curr}$ at 75 m, corresponding to the strong wind, began at 10:00 (time 3). Therefore, the difference between $V_{curr}$ at 25 and 75 m was increased at the beginning of the forced period as the wind dramatically sped up and, in turn, the velocity vertical shear increased during Stage 1 (Fig. 2c, e). The velocity difference was reduced after the strengthening of the velocity shear, and the directions of $V_{curr}$ at 25 and 75 m both changed from approximately southward to northward as the wind changed from northerly to southerly after the arrival of the typhoon eye center (Fig. 2c). The change of the velocity shear

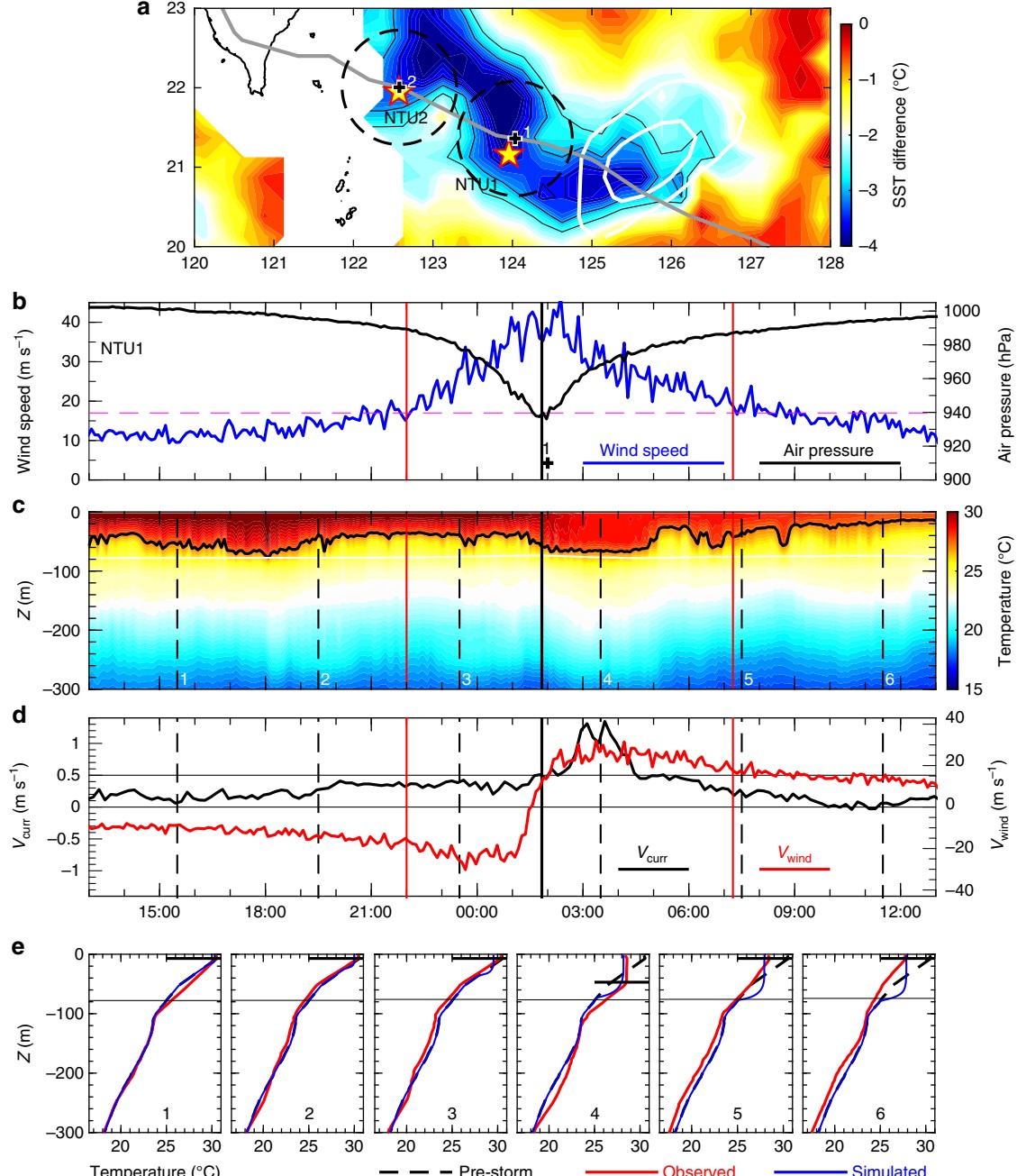

**Fig. 1** Satellite and buoy NTU1 observations of Nepartak. **a** Satellite SST difference between pre-storm SST (5 July) and post-storm SST (8 July) with the typhoon track indicated by the gray line. Dashed circles and white contours in **a** indicate the radius of the Beaufort scale 10 wind and sea level anomalies of −0.1 and −0.2 m. Buoy NTU1 observed parameters: **b** air pressure (black line) and wind speed (blue line); **c** temperature in the upper 300 m; **d** meridional current velocity $V_{curr}$ at 75 m (black line) and meridional wind velocity $V_{wind}$ (red line). The two red vertical lines in **b**–**d** mark the beginning (left) and end (right) of the forced period. The black curve in **c** is the 27.2 °C isotherm. **e** Observed (red line), simulated (blue line), and pre-storm (black dashed line) temperature profiles at times 1–6 as indicated by the vertical black dashed lines in **c** and **d**. Black lines in **e** mark the mixed layer depth. The black vertical line in **b**–**d** marks the time of minimum sea level pressure

structure potentially enhanced the turbulence and vertical mixing of the momentum and could possibly transfer the influence of wind stress to the deeper layer as seen from the depression of the 27.2 °C isotherm at the beginning of Stage 2 (Fig. 2b).

The velocity shear square $\overline{S^2}$ and the buoyancy frequency square $\overline{N^2}$ were calculated using velocity and hydrography at 25 and 75 m (Methods) and were used to examine the shear instability and its possible link to the fast temperature drop and sea surface cooling and subsurface warming observed at NTU2.

Note that the variation of density essentially follows the variation of temperature during the forced period of Nepartak. Figure 2e demonstrates that the stratification ($\overline{N^2}$) is weakened during Stage 1 (blue curve in Fig. 2e), and, at 7:00, the velocity shear increases with time (red curve in Fig. 2e) to a maximum of $7 \times 10^{-4}\,\mathrm{s^{-2}}$ at 9:40, when the stratification dramatically decreases. Thereafter, the combination of weakened stratification and strengthened velocity shear produces shear instability. The criterion for shear instability is the Richardson number $Ri$

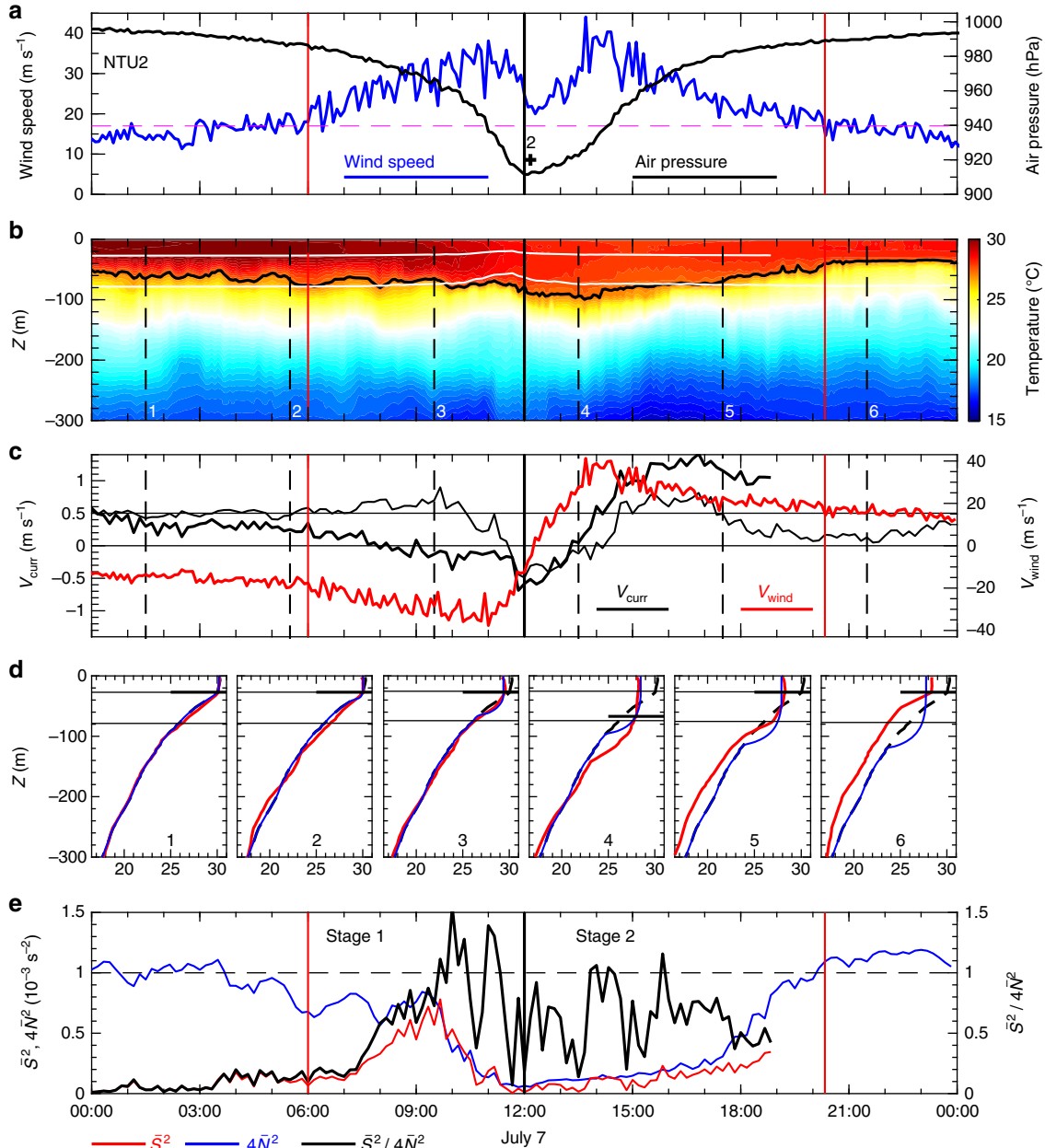

**Fig. 2** Buoy NTU2 observations of Nepartak. **a** Air pressure (black line) and wind speed (blue line); **b** temperature in the upper 300 m; **c** meridional current velocity $V_{curr}$ at 25 m (thick black line) and 75 m (thin black line), and meridional wind velocity $V_{wind}$ (red line). The two red vertical lines in **a**–**c** mark the beginning (left) and end (right) of the forced period. The black curve in **b** is the 27.2 °C isotherm. **d** Observed (red line), simulated (blue line), and pre-storm (black dashed line) temperature profiles at times 1–6 as indicated by the vertical black dashed lines in **b** and **c**. Black lines in **d** mark the mixed layer depth. The black vertical line in **a**–**c** marks the time of minimum sea level pressure. **e** Velocity shear square $\bar{S}^2$ (red line), four times of buoyancy frequency square $4\bar{N}^2$ (blue line), and $\bar{S}^2/4\bar{N}^2$ (black line). The two white lines in **b** indicate the depths of the two current meters measured by the pressure sensors of the current meter

$(=\bar{N}^2/\bar{S}^2) < 0.25$, which is transformed to $\frac{\bar{S}^2}{4\bar{N}^2} > 1$ (black curve in Fig. 2e) as the unstable condition for convenience. The value of $\frac{\bar{S}^2}{4\bar{N}^2}$ is mostly greater than 1 during the latter half of Stage 1, thereby suggesting the occurrence of shear instability and associated turbulent mixing; whereas, during Stage 2, $\frac{\bar{S}^2}{4\bar{N}^2}$ is rarely beyond the unstable condition. Our inference for the typhoon wind-induced shear instability being the primary dynamic of the rapid temperature drop during the forced period of Nepartak was examined using the one-dimensional Price, Weller, and Pinkel (PWP) model[21], which is discussed later in this study.

**Buoy data-derived air–sea heat flux and near-inertial motion.** The difficulty of concurrent atmospheric and oceanic observations during typhoons hinders the calculations of air–sea heat flux and upper ocean heat content under extreme strong wind conditions. Our observations for Nepartak provide a unique opportunity to estimate sea surface heat fluxes under these conditions. The air–sea heat fluxes are composed of the shortwave radiation ($Q_{sw}$), longwave radiation ($Q_{lw}$), sensible heat flux ($Q_{sen}$), and latent heat flux ($Q_{lat}$), which are calculated using the bulk-parameterization method[22]. The net heat flux ($Q_{net}$) is $Q_{net} = Q_{sw} + Q_{lw} + Q_{sen} + Q_{lat}$. Before the approach of Nepartak and during normal weather conditions, variations in $Q_{lw}$ and $Q_{sen}$ are

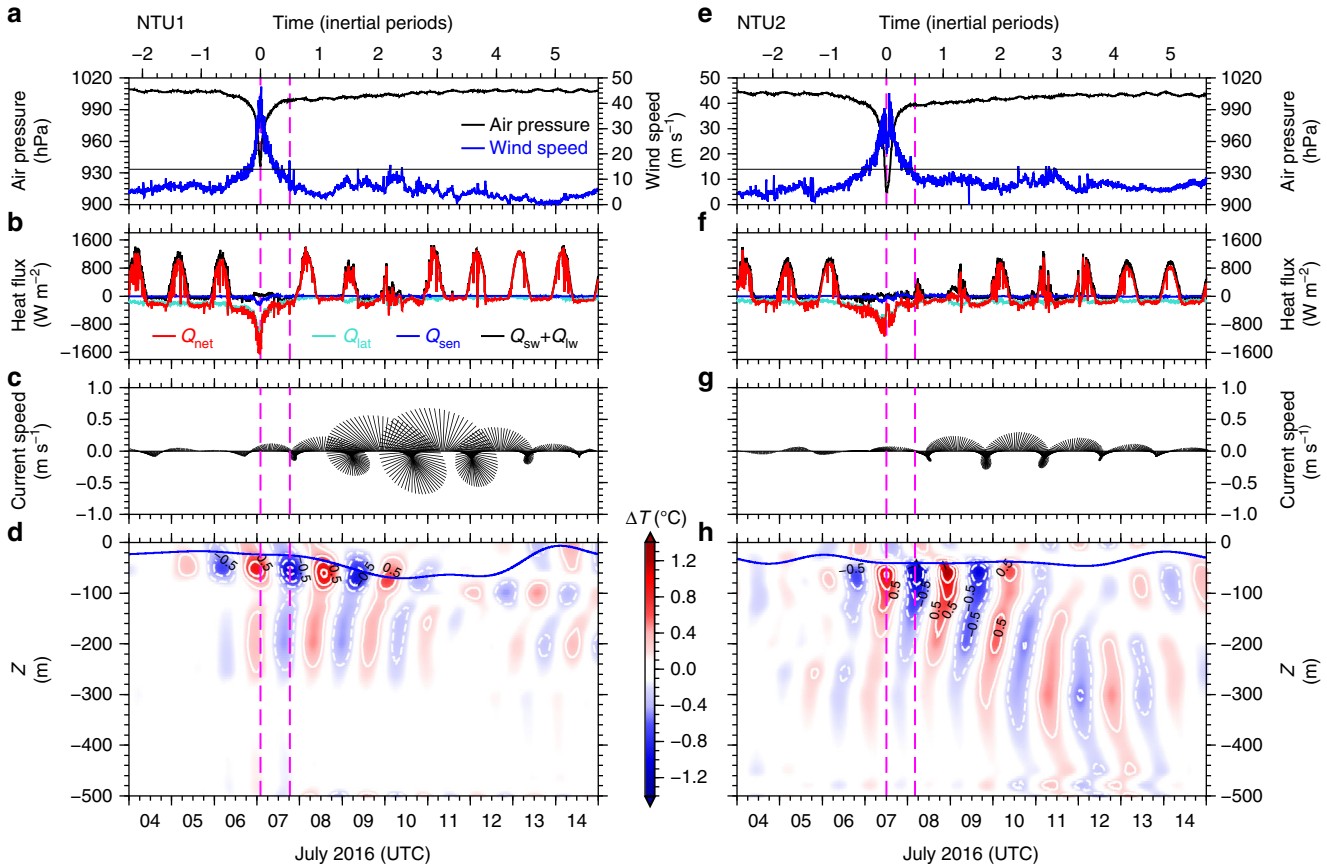

**Fig. 3** Buoy observed atmosphere and ocean variables at NTU1 and NTU2 during 4–14 July 2016. **a, e** Air pressure (black line) and wind speed (blue line), with the top axis indicates time in inertial periods and the wind speed 13.9 m s$^{-1}$ (equivalent to 50 km h$^{-1}$ or a Beaufort scale 7) indicated by the black horizontal line. **b, f** Buoy data-derived net air–sea heat flux ($Q_{net}$), net radiation ($Q_{sw} + Q_{lw}$), latent heat flux ($Q_{lat}$), and sensible heat flux ($Q_{sen}$). **c, g** Observed velocity-derived 75 m depth inertial-band (periods from 28.07 to 49.3 h) velocity sticks. **d, h** Inertial-band temperature anomalies in the upper 500 m (color shading) and the mixed layer depth (blue lines). The two magenta vertical dashed lines indicate the time at a minimum air pressure (left) and after the 0.5 local inertial period (right)

small compared with those in $Q_{sw}$ and $Q_{lat}$ (Fig. 3b, f). During the extreme strong winds of Nepartak, $Q_{net}$ reaches −1.6 kW m$^{-2}$ at NTU1 and −1.0 kW m$^{-2}$ at NTU2 (Fig. 3a, e), thus suggesting that the atmosphere gains considerable heat from the ocean. Obviously, $Q_{lat}$ dominates the heat transfer from the ocean to the atmosphere during the typhoon, which is more than 80% of $Q_{net}$. The rapid changes in $Q_{lat}$ and $Q_{sen}$ during the period from 9:00 to 15:00 7 July at NTU2 correspond to the period when the typhoon center is close to the buoy (Fig. 3e, f). The magnitudes of heat fluxes derived from the buoy data, although not surprising, are helpful in validating the numerical simulation for Nepartak. The on-site images (Supplementary Movie 1) also show that the air and sea interface becomes foggy and blurred, thus forming a two-phase transitional layer; this layer creates difficulties in determining the air temperature, wind speed, and specific humidity, etc., at 10 m above the sea surface. Since a portion of the wind momentum is used to speed up the sea spray above the sea surface, the drag coefficient can be increased slowly[23,24]. Therefore, the uncertainty in calculating $Q_{sen}$ and $Q_{lat}$ is increased because the change in the drag coefficient during extreme strong wind events is still not well-understood under disrupted air–sea interface conditions.

Inertial motions, which are generated by typhoon wind stress, wind stress curl, and wind stress convergence, particularly on the right of the typhoon track, serve as an efficient physical process in

the ocean to redistribute and dissipate the energy input from the typhoon to the ocean[25,26]. The novelty of our buoy observations is that the generation of inertial motion near the centers of super typhoons has rarely been directly observed at high temporal and spatial resolutions, although it has been occasionally observed by other data buoys[3,27–31].

The local inertial frequency ($f_0$) is $5.2580 \times 10^{-5}$ and $5.4517 \times 10^{-5}$ s$^{-1}$, corresponding to an inertial period (IP) of 33.1940 and 32.0141 h at NTU1 and NTU2, respectively. To quantify the range of direct wind influence, the radius of Nepartak during Beaufort scale 7 wind speeds, 200 km, is defined as one radius of the typhoon winds here (1 Rw). In addition to the air–sea fluxes, Fig. 3 also shows the air pressure, wind speed, inertial-band ocean current velocity at 75 m depth and the inertial motion-induced temperature anomaly ($\Delta T$) in the upper 500 m at NTU1 and NTU2. Despite the similarity of our observations and those reported in the literature, Fig. 3d, h demonstrates that significant inertial oscillations in the temperature ($\Delta T > 0.2$ °C) occur from $t = -1.25$ IP (negative or positive indicates before or after the passing of the typhoon center, respectively). At that time, the radius of the Beaufort scale 7 wind speeds was 7-Rw and 4.5-Rw away from NTU1 and NTU2, respectively, thus suggesting that the direct typhoon wind did not reach the two buoys. As the typhoon wind touched the two buoys, significant temperature anomalies extended quickly from the thermocline to ~200 m

(Fig. 3d, h). However, the development of these temperature anomalies varies differently at the two buoys. The temperature anomaly is smaller at NTU1 than at NTU2, but the strength of inertial currents is larger at NTU1 than at NTU2. The difference in the magnitude of the temperature anomaly is presumably due to the difference in the magnitude of inertial currents. The larger current-induced energetic shear instability at NTU1 decreases the temperature variation in the upper water column more at NTU1 than at NTU2, thus leading to the inertial motion-induced temperature anomaly being smaller at NTU1 than at NTU2 (Fig. 3d vs. 3h). Our observations, which are likely not observed by the other observations referred herein, provide concurrent temperature and velocity data to support this inference.

Figure 3h suggests that the temperature anomaly phase at NTU2 propagates upward and the energy propagates downward; these results are also different from those of NTU1 in Fig. 3d. The disparity in the rotation of the inertial current at NTU2 (cf. Fig. 3g, c), which is longer in time for the northward and northeastward currents than for the southwestward and southeastward currents, is presumably due to the northward Kuroshio current and on the western edge of an anticyclonic eddy (Supplementary Fig. 1b).

Furthermore, the frequency of the inertial oscillations derived from the observed velocity at NTU1 (Methods) is $5.2299 \times 10^{-5}$ s$^{-1}$, which is 0.53% smaller than the $f_0$ at NTU1. The estimated near-inertial motion frequency at NTU2 is $5.2374 \times 10^{-5}$ s$^{-1}$, which is 3.29% smaller than the local $f_0$. This discrepancy in the two frequencies is called a redshift[25], and it is consistent with some of the previous observations[29–31]. The redshift of the near-inertial motion frequency from our velocity observations is likely caused by the influence of ambient relative vorticity (Supplementary Fig. 1d) and a nonlinear effect in the upper ocean response to Nepartak's winds. The ambient relative vorticity ($\zeta$) calculated from the satellite SSH-derived absolute geostrophic current (Data) is $-0.22 \times 10^{-5}$ and $-0.08 \times 10^{-5}$ s$^{-1}$ at NTU1 and NTU2, respectively; these values may contribute to the inertial frequency shift via $f_{\mathrm{eff}} = f_0 + \zeta/2$ [32]. With this relationship, the frequency shift should be $-13.3\%$ of $f_0$ at NTU1 and $-4.8\%$ of $f_0$ at NTU2, and, apparently, the influence of the ambient relative vorticity on the local $f_0$ is better explained at NTU2 than at NTU1. The nonlinear effect raised by the similarity between the typhoon translation speed (~3.89 m s$^{-1}$) and the phase speed of the first baroclinic mode as well as the vertical variation of the ambient relative vorticity may further modify the frequency of near-inertial motion at NTU1; this phenomenon is worthy of further study.

## Discussion
In our observations, significant inertial motions were observed 0.5 IP before the arrival of the direct typhoon wind at our buoys, when the circle of Beaufort scale 7 wind speed was still 300–600 km away from the two buoys. A plausible explanation for the occurrence of the inertial motion at the two sites before the direct wind influence is that the remote effect, which comes from the fast-propagating barotropic response of the ocean to the negative air pressure anomaly of the super typhoon, induces these early arrived inertial oscillations[33]. The underlying process has not yet been examined. After the passage of the typhoon center, the amplitude of the temperature anomalies reaches a maximum during the first IP. At NTU1, the maximum inertial-band temperature variation is located right below the thermocline and varies with the deepening of mixed layer depth during the first 2.5 IPs. At NTU2, the inertial-band temperature variation is opposite in the mixed layer and below the thermocline, thus presenting a modal structure. The phase of the temperature anomaly below the

mixed layer propagates upward from ~300 m to the thermocline within the first 4 IPs. At both sites, the energy of the temperature anomaly below the thermocline tends to propagate downward. The deepening and recovery of the mixed layer at NTU1 are more noticeable than that at NTU2, which is presumably due to a stronger shear of inertial currents at NTU1 than at NTU2[28]. The modal structure in the inertial-band temperature anomaly is similar to that observed during Typhoon Kalmaegi in 2014[10] but with more spatial and temporal details present. Our observations also suggest that the nodal point varies roughly with the variation of mixed layer depth.

We focus on the detailed process and underlying dynamics for the rapid upper ocean response in the deep water region during the direct wind effects during Nepartak. The PWP model examined the evolution of the mixed layer with the initial and boundary conditions extracted from the buoy observations during Nepartak (Methods). Figures 4a and 5a demonstrate time series data of simulated temperature profiles in the upper 150 m for NTU1 and NTU2, respectively. Compared to the observations (Figs. 1c and 2b), our simulations capture essential temporal and spatial variations of the observed temperature during the forced period, thereby justifying the capability of the PWP model for resolving the upper ocean response to the extremely strong winds of Nepartak. The upper 25 m-averaged temperatures obtained from the observations ($T_{\mathrm{obs}}$) and the simulations ($T_{\mathrm{pwp1}}$) are compared. The comparison shows that the variability of $T_{\mathrm{obs}}$ (black curves in Figs. 4b and 5b) is consistent with that of $T_{\mathrm{pwp1}}$ at NTU2 (red curve in Fig. 5b) but is ~1 h earlier than the variation of $T_{\mathrm{pwp1}}$ at NTU1 (red curve in Fig. 4b); this is presumably due to the stronger advection from the confluent flow in the eddy pair at NTU1 compared to NTU2 (Supplementary Fig. 1a). The upper 25 m-averaged temperature from a similar model simulation but with zero sea surface heat flux in the model ($T_{\mathrm{pwp2}}$ in Figs. 4b and 5b) presented little difference (<0.2 °C) from $T_{\mathrm{pwp1}}$, thus verifying that the cooling effect of sea surface heat loss is very minor and the shear instability was likely the dominant mechanism causing the upper ocean temperature variation during the forced period of Nepartak.

The stability and mixing of the water column in the PWP model are determined by the criterion of shear instability, i.e., a strong vertical mixing may occur as $Ri < 0.25$ (Methods). Therefore, by evaluating the shear square ($S^2$) and the buoyancy frequency square ($N^2$) calculated from the model simulations (Figs. 4c, d and 5c, d), we are able to obtain an insight into the dynamics of the rapid temperature drop during the forced period. The high $S^2$ layer (red layer in Figs. 4c and 5c), which is approximately where the large velocity difference between the wind stress forced mixed layer and unforced lower layer occurs, deepens with the deepening of the mixed layer velocity during Stage 1 and remains almost at a constant depth during Stage 2. The high $N^2$ layer (Figs. 4d and 5d) is consistent with the high $S^2$ layer; thus, the $N^2$ layer is approximately the interface between the mixed layer and its lower layer. The shear instability occurs above the bottom of the mixed layer between 10 and 30 m as indicated by the contour of $S^2/4N^2 = 0.7$ (white curves in Figs. 4e and 5e). Additionally, the region of $S^2/4N^2 > 0.9$ (red region in Figs. 4e and 5e) coincides with the period when the mixed layer deepens, thereby lending support to our dynamical inference based on the observations. At NTU2, the occurrence of $Ri$-dependent shear instability, determined by the velocity difference between 25 and 75 m and its associated hydrography in the latter half of Stage 1 (Fig. 2e), is consistent with the period of $S^2/4N^2 > 0.9$ in our model simulations, which also occurs primarily between 25 and 75 m in the model (two horizontal white lines in Fig. 5e). Note that, as the development of the mixed layer from ~30 m to deeper than 80 m occurs quickly, the two current meters

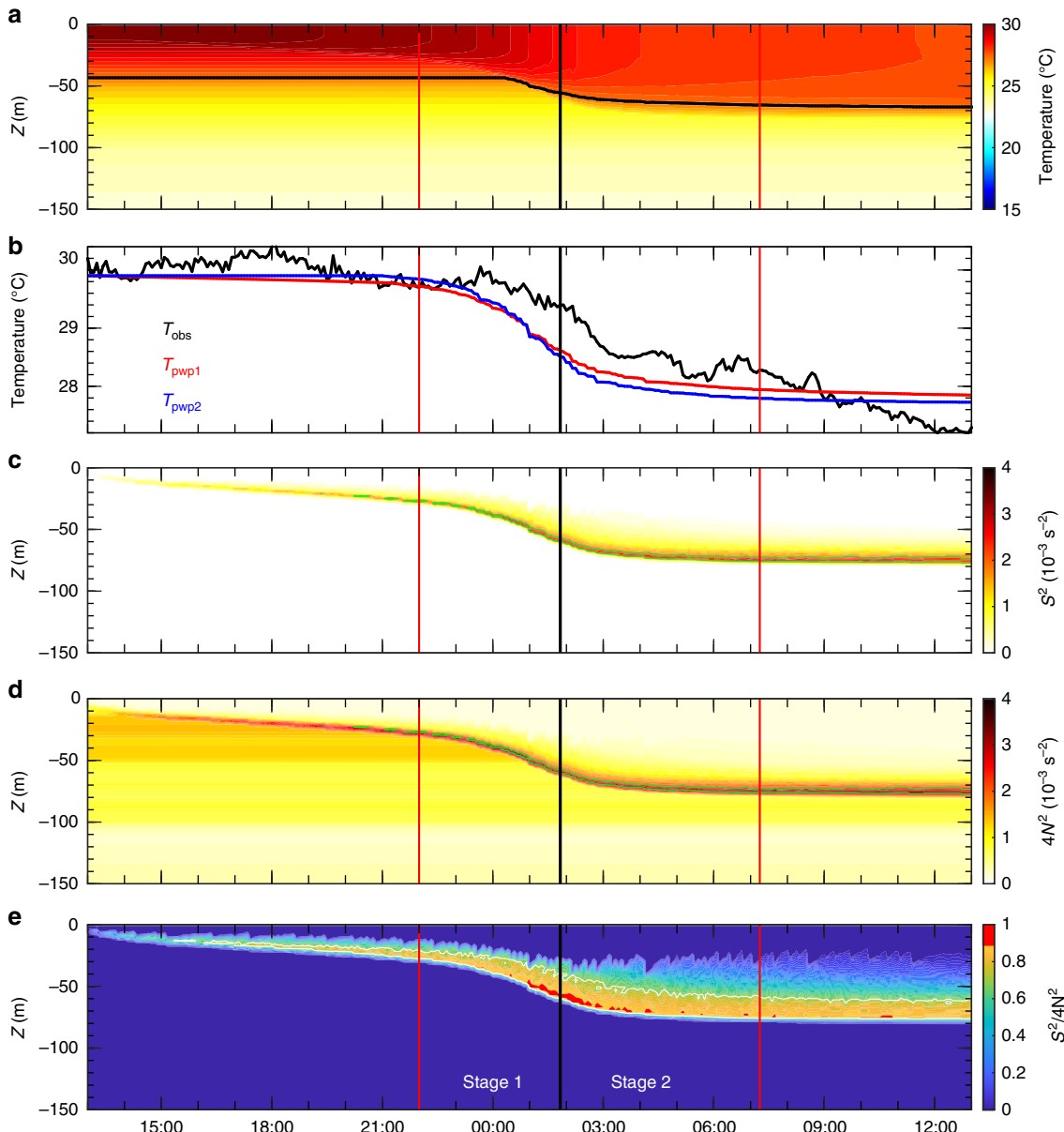

**Fig. 4** Results from the PWP model and a comparison of observational and modeling results at NTU1. **a** Model-produced temperatures. **b** Time series of the upper 25 m-averaged temperatures obtained from the observations ($T_{obs}$), from the PWP model-produced temperature ($T_{pwp1}$), and from similar model settings but with a zero surface heat flux-produced temperature ($T_{pwp2}$). Model data-derived parameters: **c** shear square $S^2$, **d** four times of buoyancy frequency square $4N^2$, and **e** $S^2/4N^2$

fixed at 25 and 75 m cannot resolve the critical velocity shear above and below the thermocline. In the model simulation for NTU2, the shear instability is still noticeable at depths deeper than 75 m during Stage 2 (Fig. 5e). This model result somewhat compensates for the limitation of our current meter observations at fixed depths and lends support to our speculation for the deepening of the shear instability and the mixed layer during the forced stage of Nepartak.

The simulated temperature profiles agree well with the observations at NTU2 (cf. blue and red curves in Fig. 2e), particularly at times 1–3. At time 4 (Fig. 2d), the mixed layer depths in the simulation and the observation are identical, except for where the observed temperature profile presents warming in the thermocline. At NTU1, the resemblance of the simulated and observed temperature profiles is reduced (cf. red and blue curves in Fig. 1e). The observed temperature profiles at times 5 and 6 are also ~1 °C

lower than the modeled temperature in deeper layers, which is likely irrelevant to the mixing and the air–sea heat exchange in the surface layer. This deep water column cooling during the forced stage is hypothetically attributed to the Ekman suction[10], which cannot be resolved by the PWP model with the settings used in this study. The Ekman suction can pump cold deep water upward to where the downward surface mixing process cannot reach.

Summarizing, our buoys observed the detailed evolution of temperature and velocity in the upper ocean under the influence of extremely strong winds (>45 m s$^{-1}$) of Super Typhoon Nepartak. The uniqueness of this dataset is that both buoys in the deep water recorded the air and sea variables when the eye of Nepartak, with its strength and translation speed remaining unchanged, passed by the buoys within a distance of 19.2 and 5.9 km to NTU1 and NTU2, respectively. This unprecedented dataset

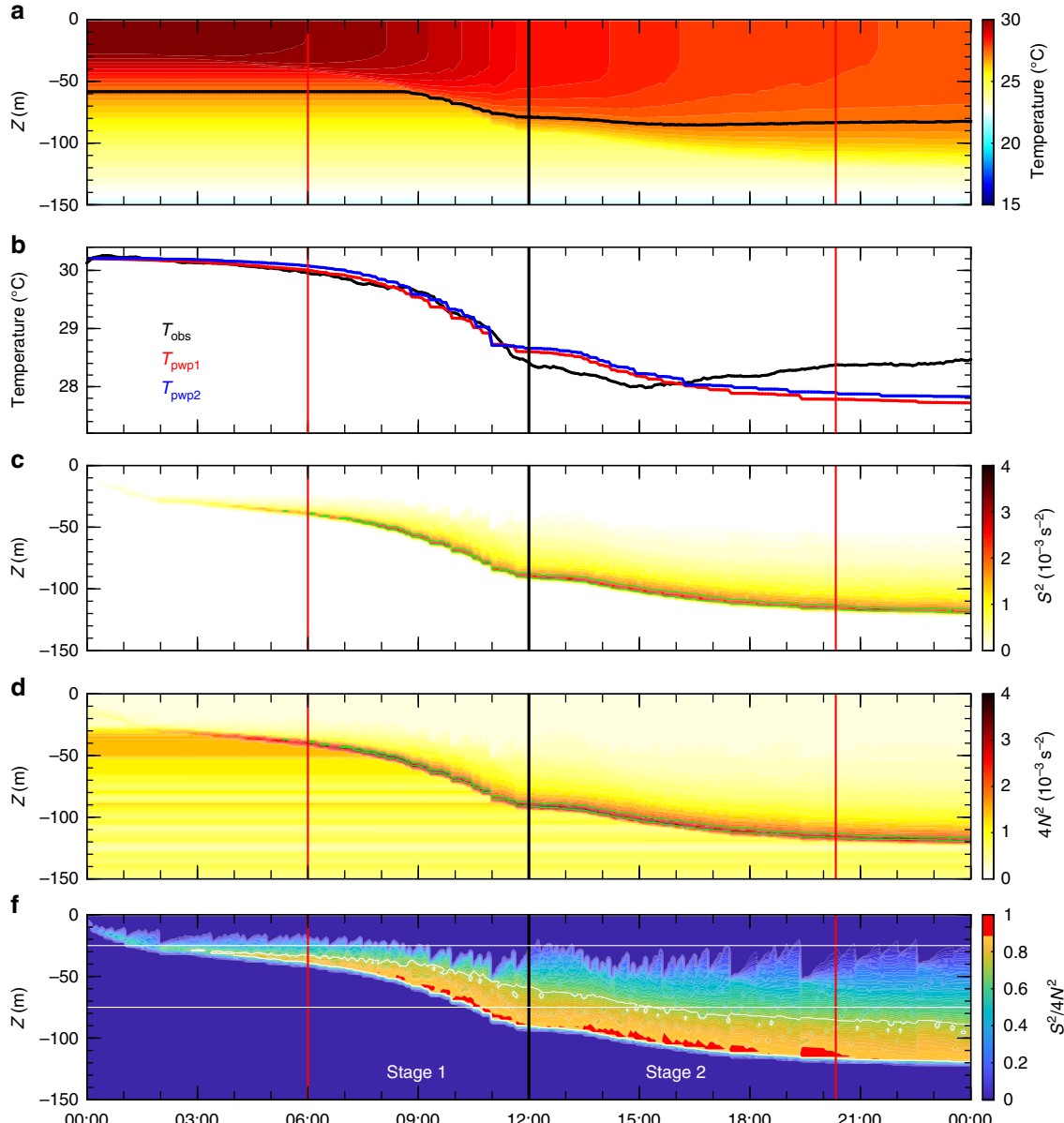

**Fig. 5** Results from the PWP model and a comparison of observational and modeling results at NTU2. **a** Model-produced temperatures. **b** Time series of the upper 25 m-averaged temperatures obtained from the observations ($T_{obs}$), from the PWP model-produced temperature ($T_{pwp1}$), and from similar model settings but with a zero surface heat flux-produced temperature ($T_{pwp2}$). Model data-derived parameters: **c** shear square $S^2$, **d** four times of buoyancy frequency square $4N^2$, and **e** $S^2/4N^2$. The two white lines in **e** mark the depths of the two current meters

provides new insights into the upper ocean responses to a super typhoon very near the typhoon's track, as well as in situ data validating numerical forecasts of Nepartak and theories of the underlying dynamics. Supplemented with numerical simulations using the PWP model, the extreme strong wind increases the velocity in the mixed layer and thus increases the velocity shear between the mixed layer and wind influenced ineffective lower layer, thereby causing shear instability during the first stage of the forced period. This shear instability in turn produces strong turbulence mixing, leading to an ~1.5 °C temperature drop in 4 h and a cooling–warming pattern of temperature variation in the upper ocean. The turbulence mixing deepens the mixed layer and rapidly transfers the momentum input from the surface wind stress to ~120 m. The two buoys also observed a deep water cooling within the bottom of the mixed layer and >300 m depth during the direct wind influence of Nepartak, which is potentially

caused by Ekman suction. Although the upper ocean temperature decreased quickly when the direct wind of Nepartak was still a major influence, the strength of Nepartak was not influenced as it passed by the two buoys. The buoy observations also detail the development and evolution of the inertial current and temperature oscillations occurring before the arrival of direct typhoon wind at the two buoys, and a pronounced redshift in the frequency of inertial motion, particularly at NTU1.

## Methods

**Buoy configuration**. The oceanic and atmospheric data were collected by moored surface buoys with improved low power consumption data acquisition systems, new electric power schemes, inductive modems, fishing line cutters, and satellite communications modules[9]. The solar panels that supply electric power to the system were replaced by a tube-like lithium battery pack, which largely reduced the drag of the stainless steel tower on the buoy during extremely high typhoon winds. In June 2016, two buoys were deployed at 123.9° E, 21.1° N (station NTU1) and

122.6° E, 21.9° N (station NTU2), which are 375 and 175 km, respectively, southeast of the southernmost point of Taiwan (Supplementary Fig. 1a). The water depths at these two locations are 5483 and 4877 m, respectively. Each buoy measures wind, air temperature, air pressure, humidity, precipitation, solar radiation, and irradiance above the sea surface (Supplementary Fig. 2); high-resolution ocean temperature and salinity in the upper 500 m and at fixed depths 700 and 900 m, and ocean current velocity at 25 and 75 m depths (Supplementary Fig. 3).

**Buoy observations**. The sampling interval is 6 min for atmospheric sensors and the sea surface temperature probe, 1 min for conductivity–temperature–depth and temperature–pressure sensors under the sea surface, and 10 min for the two current meters. The data can be transmitted at time intervals of 30 min during regular weather or 6 min during typhoons, via Iridium satellite communication. The data were also stored in separate data loggers, which can alternatively be retrieved through radio communication between the buoy and a nearby ship.

Typhoon Nepartak was a tropical storm south of Guam that occurred on July 3 and became a super typhoon on 6 July with a well-developed, distinguishable eye[11]. The two buoys observed Nepartak from where they were moored, and the closest distance from the typhoon's center to the two buoy locations was 19.2 and 5.9 km for NTU1 and NTU2, respectively. The raw data were transmitted via satellite to the ground station in real-time, and they were retrieved later from the data loggers after the recovery of the two buoys in October 2016. The coordinate of Nepartak's eye, provided by Taiwan's Central Weather Bureau, was determined by the satellite imagery of Himawari-8 with 0.1 × 0.1° resolution (~11 km at 20° N) every 3 and 1 h when Nepartak was over the sea and land, respectively. The maximum error in the track estimation is thus ~15 km.

**One-dimensional mixed layer model**. The PWP model[21] is a one-dimension numerical model with the static stability, bulk Richardson number ($R_b$), and gradient Richardson number ($R_g$) as criteria to determine the stability and, in turn, the turbulent mixing in the water column. The model solves the momentum equation to obtain velocity, temperature, and salinity under user specified initial hydrographic conditions and boundary conditions of wind stress (momentum transfer), heat, and salinity fluxes. During the model calculation, criteria of the static stability $\frac{\partial \rho}{\partial z} \leq 0$ and mixed layer stability $R_b = \frac{-gh\Delta\rho}{\rho_0(\Delta v)^2} \geq 0.65$ ($\Delta\rho$: density difference between the mixed layer and the level just beneath the mixed layer; $\Delta v$: similar to the definition of $\Delta\rho$ but for velocity; $h$: the mixed layer thickness) are checked throughout the water column, and shear flow stability $R_g = \frac{-\frac{g}{\rho_0}\frac{\partial\rho}{\partial z}}{(\partial v/\partial z)^2} \geq 0.25$ is applied to stratified levels below the mixed layer. If any of the former two criteria is not satisfied, densities of adjacent vertical levels are artificially mixed in the model. If the criterion of shear flow stability is violated, the water column is partially mixed until the entire water column satisfies this criterion.

In the simulation of the upper ocean response to Nepartak at the two buoy locations, the thickness of water column was set at 500 m with a vertical level resolution of 2 m in the model. The initial temperature profile was determined by the buoy observations before the forced period, i.e., black dashed lines in Figs. 1e and 2d, and the salinity was fixed at 34.5 throughout the simulation for simplicity. The observed wind of Nepartak, i.e., blue curves in Figs. 1b and 2a, was used to estimate the wind stress[34] used in the model. The net sea surface heat flux calculated from the observations, i.e., $Q_{net}$ in Fig. 3b, f, was used as the sea surface boundary condition. The surface salinity flux was set at zero. The model run with these settings except that the sea surface heat flux was set to be zero was also conducted for evaluating the importance of air–sea heat exchange in the upper ocean response to extreme strong typhoon wind.

**Calculation of velocity shear and buoyancy frequency**. The velocity shear square is computed as $\overline{S}^2 = (\Delta u/\Delta z)^2 + (\Delta v/\Delta z)^2$, where $\Delta u$ and $\Delta v$ are the velocity differences between 25 and 75 m in the zonal and meridional directions, respectively. The buoyancy frequency square is computed as $\overline{N}^2 = -\frac{g}{\rho_0}\frac{\Delta\rho}{\Delta z}$, where $\Delta\rho$ is the density difference between 25 and 75 m, $\rho_0$ (=1025 kg m$^{-3}$) is a reference density, and $g$ (=9.81 m s$^{-2}$) is the gravitational acceleration.

**Subtraction of tidal currents**. The velocity component of tidal currents in the observed raw velocity data was extracted using harmonic analysis. Harmonic constants of five dominant principal constituents ($O_1$, $K_1$, $N_2$, $M_2$, and $S_2$), obtained using T_Tide software[16], were used to compose the tidal currents. The composed tidal currents were then subtracted from the observed raw velocity, obtaining de-tided current velocity. A similar procedure was applied to the temperature records to obtain de-tided temperatures.

**Mixed layer depth**. The mixed layer depth was determined as the depth at which the temperature difference exceeds 0.5 °C within a 1 m interval.

## Data availability
The quality assured/controlled buoy data are available online at https://po.oc.ntu.edu.tw/buoy/manage/login.php. The satellite SSH-derived absolute geostrophic current is distributed by the Archiving, Validation, and Interpretation of Satellite Data in Oceanography (AVISO), which can be downloaded at AVISO via Ssalto/Duacs (http://www.aviso.oceanobs.com/en/data/products.html). The satellite sea surface temperature is the product of the microwave Optimally Interpolated (OI) SST downloaded at http://data.remss.com/SST/daily/.

## Code availability
The MATLAB code of PWP model is available through https://github.com/OceanMixingGroup/mixingsoftware/tree/master/pwp.

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

## Acknowledgements

The data buoy observation project was supported by the National Taiwan University (NTU) under the Aim for the Top University Plan, the Institute of Oceanography, NTU, the Ministry of Science and Technology (MOST), and Central Weather Bureau, Taiwan. Y.J.Y. was supported by the MOST grant 105-2611-M-002-014. M.-H.C. was supported by the MOST grant 105-2611-M-002-012. S.J. was supported by the MOST grant 105-2119-M-002-042. H.-I.C. was supported by the MOST grant 104-2611-M-002-012-MY2. The technicians at the Marine Instrument Center of NTU and captains and crew of R/Vs Ocean Researcher I and III helped deploy, maintain, and recover the buoys. Tien-Hsia Kuo prepared Supplementary Figure 1 and Cheng-Chia Lien produced the Supplementary Movie. The staff at the Department of Atmospheric Sciences, NTU helped calibrate the barometers. The MATLAB code of PWP model used in this study is maintained by Dr. Sally Warner.

## Author contributions

Y.J.Y. conducted the design and led the fieldwork of the deployment/recovery of the data buoys. Y.J.Y., M.-H.C., S.J., and C.-Y.H. analyzed the data and performed relevant figures. Y.J.Y., M.-H.C., and S.J. wrote the initial draft. Y.J.Y. and M.-H.C. contributed to providing the fundamental idea for interpreting the data. H.-I.C. designed the electronic control system of the buoy. C.-L.W. led the project and overall coordination. All authors contributed to the final text and figures.

## Additional information

**Competing interests:** The authors declare no competing interests.

