## [Peer Review File · Nature Communications]

Reviewer #1 (Remarks to the Author):

Review of:

New insights of heat flux and velocity shear in ocean responses to Super Typhoon Nepartak.

by Yang, Chang, Hsieh, Chang, Jan & Wei.

General comments:

This manuscript presents met-ocean observations during Super Typhoon Nepartak as it rapidly tracked across two improved NTU moorings deployed east and southeast of Taiwan. Each of the moorings included a surface buoy instrumented with a suite of meteorological sensors, a high resolution upper ocean CTD/TD string, and current meters at two depths. Peak winds observed by the buoys in Nepartak look to be about 40 m/sec, and as such have the potential to provide unique insights on the ocean response during extreme forcing.

The present manuscript (a) describes the rapid changes in the upper ocean temperature field that occur during the direct forcing phase of Nepartak, (b) uses the TOGA COARE bulk formulations and the current meters at 25 m and 75 m depth to begin looking at the relative influence and timing of surface heat fluxes and shear induced mixing on the surface layer cooling during the important direct forcing period, and (c) present results of an analysis of the inertial velocity and temperature response starting just before the storm and running through the inertial tail.

The authors can be commended for the survival of their mooring design in extreme conditions and buoy placement based on historical typhoon tracks as outlined in the supplemental materials. No doubt that the mooring observations in Nepartak will lead to publishable results, but there is still work to do, especially to meet the Criteria for Publication in Nature. In general, important details of the rapid temperature change during the direct forcing period are left out and cannot be deciphered using the existing figures. The analysis of the vertical shear is based on current meters fixed in space, and as the pycnocline deepens during the storm, the invaluable measure of shear across the pycnocline is transformed into a much less useful measure of the shear within the surface layer. Lastly, the description of the inertial response is not used to provide insights on the heat flux and velocity shear noted in the paper title, and the inertial response in Nepartak is not compared to many previously documented observations of the inertial tail to clearly state what is new.

The part of this work most likely to yield novel results is the relative impact of the heat fluxes and the shear-induced vertical mixing on the rapid upper ocean response during the direct forcing time period when the upper ocean can still impact the intensity of this typhoon. To get there, the existing

analyses would benefit greatly from modeling studies that move beyond speculation to providing new insights and new understanding. Even a simple modeling study with a Price-Weller-Pinkel type model would be helpful. Previous studies have already raised the bar to this level or beyond.

I offer two papers that may provide the authors insights on how to proceed. One is the Zhang et al. (2014) Typhoon Kalmaegi paper in JGR Oceans already referenced twice (#8 & #28). Second is an unreferenced paper on shear-induced mixing in hurricanes and typhoons published in Nature Communications by Glenn et al. (2016) on stratified coastal ocean interactions with tropical storms. Both papers describe new observations in typhoons and hurricanes, use models verified by the observations to understand the dynamics of the response, and further use the modeled heat budget to determine what processes (vertical mixing, horizontal or vertical advection, etc.) are responsible for the observed rapid changes in ocean temperature during the directly forced portion of the storm. A future paper on Typhoon Nepartak will greatly benefit from similar modeling studies that demonstrate the relative significance of heat fluxes and vertical shear to the rapid upper ocean response. If the broader impact motivation for this paper is greater understanding “to improve the accuracy of the numerical weather typhoon forecast” (line 35), then a focus on the direct forcing stage is relevant.

Specific comments:

1) First sentence of the abstract, I would add the word “rapid” somewhere, as in “Rapid upper ocean responses to typhoons are key to understanding...” to emphasize the significance of the Nepartak observations. The authors have successfully captured very rapid changes in upper ocean temperatures that occur on time scales of a few hours while the storm is still present. Much previous work on the ocean response to tropical storms focuses on the inertial tail after the storm has passed, and the potential for the cold wake to influence the next storm. The rapid response observed during the direct forcing period of this typhoon has the potential to feedback on typhoon intensity. Documenting these rapid ocean changes is critical to demonstrating the need for coupled atmosphere-ocean models for typhoon forecasting. The authors have collected a significant dataset. I encourage them to continue working towards publication.

2) Lines 20-23: The following statements are directly supported by the observations: “The temperature cooling decreased the density difference between the mixed layer and the lower water” and “whereas the strong wind-driven current in the mixed layer increased the velocity difference vertically”. But this is followed by the statement: “The associated velocity shear weakened the resistance of the pycnocline to the deepening of the mixed layer by producing shear instability and turbulent mixing”. Without observations or a model to support it, this potentially important result is speculation. Shear induced mixing early in the direct forcing stage is exactly the process responsible for rapid upper ocean cooling found by Glenn et al. (2016) and published in Nature Communications. The Glenn et al. (2016) paper looks at hurricanes and typhoons in shallow water that are approaching landfall. Nepartak is a potential example of shear induced mixing causing rapid cooling in deepwater, so it is worth the effort to investigate.

- 3) Line 25: Where is the discussion of the “temperature deviation developed downward to 500 m in 5 hours”? I see figure 4 plots extend to 500 m depth. What in these plots is this statement referring to? I see little difference before, during or after the storm at 500 m at short times scales, and some differences at NTU2 at 500 m a few days after the storm.

- 4) Line 26-28: Modification of the observed near-inertial period by the relative vorticity of the background flow is well established. What insights does this provide on the upper ocean response, especially during the direct forcing stage when the upper ocean can feedback on Nepartak’s intensity?

- 5) Lines 56-68: This paragraph describe the findings from similar deepwater met-ocean moorings in Hurricane Fabien, Typhoon Choi-Wan and Typhoon Kalmaegi. Perhaps a table noting the maximum wind speed observed by the mooring in each case and the closest approach of the eye center would help put the Nepartak observations in context and help support statements like “unprecedented data set” on line 81 and “the only in situ high resolution observation of a super typhoon, with its center passing near two data buoys” on lines 87-88.

- 6) Line 61. This manuscript lists Typhoon Choi-Wan as Category 5. Bond et al., 2011 (Ref #5) lists it as Category 1 on their page 3. Figure 4 in Bond et al. (2011) also shows the most significant cooling occurring during the upwelling of the thermocline just after the eye center passage. The Nepartak data is different, with significant cooling before eye center passage while the thermocline is deepening. These details in the Nepartak observations that make them different from observations in other extreme storms are difficult to explore in the existing figures.

- 7) Line 83: This paper examines the measured “ocean mixed layer” and computed “air-sea heat fluxes”. Air-sea fluxes are not measured, but are computed based on the TOGA COARE bulk algorithms. There is increased uncertainty in these algorithms at very low and very high wind speeds.

- 8) Line 88-90: How does the video contribute to the new insights on the upper ocean response? Maybe stills during specific phases of the storm can provide these insights? Some discussion of the video and how it is relevant to the interpretation of the upper ocean response is required if the video is to be included.

- 9) Lines 96 & 98. Here it says closest approach of Nepartak to the buoy NTU1 was 15 km and buoy NTU2 was 5.7 km. Abstract says 19.2 km and 5.9 km. Zhang et al. (2016) (Ref #8 & #28) also discuss the track uncertainty from different sources. Data section does not say which source for the track was used, and does not include any measure of the uncertainty in the track at the time of closest approach to the moorings.

- 10) Line 102. That the cold wake is 2.5C cooler than the surrounding water is an interesting observation. But the main purpose of this paper is the upper ocean response, especially during the direct wind forcing section. I am not sure which SST images are cloud free, but a before, after and difference SST map is almost standard for this. Also, the source of the SST data is not described in the data section.
- 11) Line 104. East of 124E, stating that the cold water on the left-hand side of the track is in the same location as the cold eddy is defensible. Saying it is "due to" the pre-existing cold eddy modulating the shape of the cold wake, a cold wake that is related to the inertial response, is unsupported speculation that could be investigated with a model.
- 12) Lines 105. A focus on the upper ocean temperature and velocity response during the forced period remains an excellent focus for any revisions of this paper. Relating this to the storm intensity would be useful. It may be as simple as noting if the intensity is remaining steady or how it is changing during this time period.
- 13) Lines 109-111. The definition for the forced period used here (start of temperature response and end of shear instability development) is not based on the forcing, but on the response. This should be reconsidered. The strong forcing can start before the temperature responds, and the change in vertical shear is also influenced by the deepening of the pycnocline. The reduction in the observed shear used to define Stage 2 is at least partially related to the pycnocline migrating below the bottom current meter so that both current meters are eventually in the upper layer with little shear observed. There still may be significant shear between the deepening surface layer and the deeper layer below, something that a model could reveal.
- 14) Lines 115-119. The details of the thermal structure described here would be greatly aided by some profile plots. Sample profiles at critical times in the storm could be shown to aid the discussion throughout.
- 15) Line 122 & 141. Stage 1 & 2 definitions during the forced period are based on the responses noted above, which have some inherent flaws, and are only definable for one of the moorings due to current meter failures. Glenn et al. (2016) also divide the forced period into two stages, ahead-of-eye-center and after-eye-center based on the air pressure record to define the eye center and the wind speed to define the start and end of the forced period. Adopting these definitions will enable the upper ocean response to be examined for both moorings during a forced period that can still be divided into two relevant stages. For example, there is a significant change in the TOGA COARE computed heat fluxes immediately after eye passage based on the air pressure record in both moorings. Time series of SST and air temperature to accompany the existing time series of air pressure and wind speed could provide further insights as to why.

16) Lines 143-146. The way to demonstrate that “turbulent mixing induced by shear instability, rather than air-sea heat flux, dominates the seawater cooling” is to use a model and look at the terms in the equations for the heat balance.

17) Line 145. N_2 is introduced here, and the process to calculate it is described in lines 416-418. But there is only the Figure 1 plot of the temperature data, with no similar plot or discussion of the salinity or density field. It is understood that Nature figures are supposed to be compact, but even a short description of the salinity or density time series, or salinity and density plots similar to 1b and 1g in the supplemental materials, would help.

18) Lines 153-155. As noted above, there are multiple reasons why the 4N2-S2 is small during Stage 2. Early on in the storm, the 25 m and 75 m observation depths are on different sides on the pycnocline. As the pycnocline deepens during the storm, the observation depths are increasingly both in the upper layer. This is a fatal flaw when trying to use just the observations to describe the processes. Using the observations to validate a model that is then used for interpretation of processes is a more likely to be successful.

19) Lines 158-178. Interpretation of Figure 2 supports the above alternate interpretation of the current meter record. During Stage 1, the currents are in difference directions, and the temperatures differ by about 4C. During Stage 2, currents are in the same direction and temperature difference quickly goes to zero, implying the 2 current meters are in the same layer.

20) Line 183. What does the depth variation of the current meter (gray line) mean? Is this a measure of the waves?

21) Line 217. Caption says “observed” heat flux. It is computed based on TOGA COARE.

22) Lines 221-226. Is the comparison of the computed air-sea fluxes from the buoy with the time series derived from the OAF flux maps necessary? The OAF flux maps are not expected to resolve the storm, and this shows they do not. How does this result provide new insights into the upper ocean response? It does not seem necessary. It is certainly not a conclusion that one would publish in Nature.

23) Lines 230-235 explain some uncertainties in the TOGA COARE algorithms, that they may not apply at high windspeeds, which is why they are not direct observations of the heat flux. Some discussion of this uncertainty will help show that the general relative magnitudes and trends in the fluxes are still valid.

- 24) Lines 236-299. Inertial response of the ocean to typhoons is well documented as noted by the authors in references 3, 21, 22, 23, 24, and 25. What new insights does this provide, and how does it contribute to the understanding of heat flux and velocity shear impacts on the ocean response?
- 25) Line 276 (and others): The discussion refers to time in reference to inertial periods, but the time axis in the figures is labeled in days during the month of July. This makes it difficult to use the figure to understand and verify the discussion points. Can the inertial period time line be added to these figures to better follow the discussion?
- 26) Lines 306-308. This is an important conclusion, that the vigorous mixing leads to the rapid decrease in sst during the direct forcing stage while the storm is still present, and that negative feedback can weaken the storm. Focusing on this result, and comparing it to previous observations and modeling studies has the potential for new insights. For example, for tropical storms in deepwater, the Price models indicate that the rapid cooling during the direct forcing period is expected to be approximately evenly split between ahead-of-eye-center cooling and after-eye-center cooling. Is this what is observed in Nepartak? It is difficult to tell from Figure 1 alone.
- 27) Lines 315-317. A model result would demonstrate this.
- 28) Lines 318-320. This process of the surface layer accelerating in response to the wind resulting in vertical shear between the surface and deep bottom layer that then produces enhanced mixing is a known process in the Price models.
- 29) Lines 324-325: Not sure where the temperature variations developing downward to 500m within 5 hours is discussed.
- 30) Lines 332-334: I would expect that the maximum inertial band temperature variation below the deepening thermocline is the oscillation of the thermocline depth as has been observed in the inertial tail of other typhoons. But the blue line in Figure 4c and 4f for the mixed layer depth does not oscillate. Is the blue lined smoothed?
- 31) Lines 343-355: This uses geostrophic currents from satellite derived sea surface height to calculate the background relative vorticity to see if this is responsible for the shift in the observed near-inertial frequency. Good agreement is achieved with NTU2, which, based on supplemental figure S1, is located between the Kuroshio and the anticyclonic mesoscale eddy. Agreement is not as good with NTU1, which is located between two mesoscale eddies of opposite signs. While Figure S1 is not sufficient to tell the exact locations of the buoys relative to these ocean features, it seems that errors in the location of the two eddies can have a large impact on the background vorticity estimate at NTU1, even changing the sign of the relative vorticity. This could be investigated as potential

explanation of the discrepancy between the observed near inertial frequency at NTU1 and the frequency estimated here. A map of the geostrophic current fields or sea surface height contours with the buoy locations properly marked would be a first step, and is easily included in the supplemental materials.

Minor typos:

Line 35, "forecast" should be "forecasts".

Line 37, what are "false announcements of the typhoon day"? The forecast landfall day?

Line 42, "inertia" should be "inertial".

Line 133, air pressure is light blue and wind speed is orange.

Line 138, what are the white lines in Figure 1e?

Line 243. Units for inertial frequencies are usually 1/seconds.

Line 250. Notation of " $t = -1.25 \cdot IP$ " is confusing. I interpreted this as $t = -1.25 \cdot IP$

Line 509. "white" line.

Reviewer #2 (Remarks to the Author):

This paper describes some very useful observations made a new kind of deep sea data buoy that is able to survive encounters with very intense storms. The data are therefore quite valuable, and I do believe that this paper is worthy of publication in a very high quality international journal. I am guessing that the editors will ask that the paper be made more concise, and I encourage that as well; omit everything that does not directly aid the reader.

Some details regards the discussion.

1) Suggest that you check to see how big the temperature change due to the surface heat flux should be (evaluate a 1-d heat budget). No doubt that the surface heat flux contributes to SST cooling, but I expect that it will not be a very large fraction of the total cooling even during the first stage.

2) Estimating the effect of shear on stability is worthwhile, but is somewhat limited since only two depths are available. Very likely the stability was lower at other depths not sampled. Nothing you can do about that other than be aware of it and take care not to over-interpret these data. Specifically, higher than critical stability between 25 and 75 m does not foreclose the possibility that stability was lower elsewhere and contributed to vertical mixing during the first stage.

3) The discussion of the deep temperature response (lines 323+) is a bit muddled. It almost sounds as if it is an extension of the cooling/mixing processes that cause SST cooling. However, the deep response is obviously reversible (it oscillates) and so is not due mainly to a mixing process. Rather, it is almost certainly associated with inertial pumping that you made reference to earlier in the paper.

4) Not sure I see the value of the horizontal shear effects on the inertial oscillation frequency. Unless I misunderstood, the estimated effect is much less than was actually observed?

Hope these will be of some help. Good luck with revisions!

Responses (in blue) to reviewer #1's comments.

Review of:

New insights of heat flux and velocity shear in ocean responses to Super Typhoon Nepartak. by Yang, Chang, Hsieh, Chang, Jan & Wei.

General comments:

This manuscript presents met-ocean observations during Super Typhoon Nepartak as it rapidly tracked across two improved NTU moorings deployed east and southeast of Taiwan. Each of the moorings included a surface buoy instrumented with a suite of meteorological sensors, a high resolution upper ocean CTD/TD string, and current meters at two depths. Peak winds observed by the buoys in Nepartak look to be about 40 m/sec, and as such have the potential to provide unique insights on the ocean response during extreme forcing.

The present manuscript (a) describes the rapid changes in the upper ocean temperature field that occur during the direct forcing phase of Nepartak, (b) uses the TOGA COARE bulk formulations and the current meters at 25 m and 75 m depth to begin looking at the relative influence and timing of surface heat fluxes and shear induced mixing on the surface layer cooling during the important direct forcing period, and (c) present results of an analysis of the inertial velocity and temperature response starting just before the storm and running through the inertial tail.

The authors can be commended for the survival of their mooring design in extreme conditions and buoy placement based on historical typhoon tracks as outlined in the supplemental materials. No doubt that the mooring observations in Nepartak will lead to publishable results, but there is still work to do, especially to meet the Criteria for Publication in *Nature*. In general, important details of the rapid temperature change during the direct forcing period are left out and cannot be deciphered using the existing figures. The analysis of the vertical shear is based on current meters fixed in space, and as the pycnocline deepens during the storm, the invaluable measure of shear across the pycnocline is transformed into a much less useful measure of the shear within the surface layer. Lastly, the description of the inertial response is not used to provide insights on the heat flux and velocity shear noted in the paper title, and the inertial response in Nepartak is not compared to many previously documented observations of the inertial tail to clearly state what is new.

The part of this work most likely to yield novel results is the relative impact of the heat fluxes and the shear-induced vertical mixing on the rapid upper ocean response during the direct forcing time period when the upper ocean can still impact the intensity of this typhoon. To get there, the existing analyses would benefit greatly from modeling studies that move beyond speculation to providing new insights and new understanding. Even a simple modeling study with a Price-Weller-Pinkel type model would be helpful. Previous studies have already raised the bar to this level or beyond.

I offer two papers that may provide the authors insights on how to proceed. One is the Zhang et al. (2014) Typhoon Kalmaegi paper in *JGR Oceans* already referenced twice (#8 & #28). Second is an unreferenced paper on shear-induced mixing in hurricanes and typhoons published in *Nature Communications* by Glenn et al. (2016) on stratified coastal ocean interactions with tropical storms. Both papers describe new observations in typhoons and hurricanes, use models verified by the observations to understand the dynamics of the response, and further use the modeled heat budget to

determine what processes (vertical mixing, horizontal or vertical advection, etc.) are responsible for the observed rapid changes in ocean temperature during the directly forced portion of the storm. A future paper on Typhoon Nepartak will greatly benefit from similar modeling studies that demonstrate the relative significance of heat fluxes and vertical shear to the rapid upper ocean response. If the broader impact motivation for this paper is greater understanding “to improve the accuracy of the numerical weather typhoon forecast” (line 35), then a focus on the direct forcing stage is relevant.

General response.

We thank the reviewer for the in-depth review and the constructive suggestions. To strengthen the findings from the observations of the two buoys, in this revision we focus on the details of the rapid cooling in the upper ocean during the forced period of Nepartak with improved figures for the observed air/sea variables as the reviewers suggested. The associated dynamics is examined using the one-dimensional Price-Weller-Pinkel model with the observations as the initial and boundary conditions. The model simulations actually provide promising support to our previous speculation for the dominant role of the typhoon-enhanced velocity shear instability in the rapid upper ocean cooling and concurrent deepening of the mixed layer. The interpretation and discussion on the calculated sea surface heat flux and near-inertial oscillation induced by Nepartak is considerably revised with the comparison with the previous published observations. The paper of Glenn et al. (*Nature Communications*, 2016) is of crucial importance for the interpretation of our data that present fast upper ocean responses near the track of Nepartak in the deep water region, and the paper has been cited here. The manuscript has been revised thoroughly in accordance with the two reviewers' comments, and we feel that the manuscript now well satisfies the journal's quality.

Specific comments:

- 1) First sentence of the abstract, I would add the word “rapid” somewhere, as in “Rapid upper ocean responses to typhoons are key to understanding...” to emphasize the significance of the Nepartak observations. The authors have successfully captured very rapid changes in upper ocean temperatures that occur on time scales of a few hours while the storm is still present. Much previous work on the ocean response to tropical storms focuses on the inertial tail after the storm has passed, and the potential for the cold wake to influence the next storm. The rapid response observed during the direct forcing period of this typhoon has the potential to feedback on typhoon intensity. Documenting these rapid ocean changes is critical to demonstrating the need for coupled atmosphere-ocean models for typhoon forecasting. The authors have collected a significant dataset. I encourage them to continue working towards publication.

Reply: The reviewer's suggestion has been thoughtfully incorporated into the revision of the manuscript. We strengthened the motivation and objective of this study, which can be regarded as a compensation and extension of the high-quality study conducted by Glenn et al. (2016). An associated statement has been added to introduction: “Previous observations in stratified coastal ocean and associated studies (Glenn et al., 2016; Seroka et al., 2017) have proven that velocity shear-induced vertical mixing during direct wind influence of tropical cyclones is the primary mechanism responsible for rapid upper ocean cooling while the storm wind is still effective. The rapid cooling caused intensity reduction of some hurricanes translated across the continental shelf of the Mid-Atlantic Bight (Glenn et al., 2016). While in deep water, whether

or not similar rapid cooling during direct forcing period of tropical cyclones remains unclear.” The abstract has been modified to fit the word limit (150) set by Nature Communications and the key term “rapid temperature drop” during the direct wind influenced period is appeared in the second sentence to highlight the significance of the observation. We also emphasize the high temporal resolution data set is crucial to the post-evaluation of the numerical forecast for Nepartak in this revision.

- 2) Lines 20-23: The following statements are directly supported by the observations: “The temperature cooling decreased the density difference between the mixed layer and the lower water” and “whereas the strong wind-driven current in the mixed layer increased the velocity difference vertically”. But this is followed by the statement: “The associated velocity shear weakened the resistance of the pycnocline to the deepening of the mixed layer by producing shear instability and turbulent mixing”. Without observations or a model to support it, this potentially important result is speculation. Shear induced mixing early in the direct forcing stage is exactly the process responsible for rapid upper ocean cooling found by Glenn et al. (2016) and published in *Nature Communications*. The Glenn et al. (2016) paper looks at hurricanes and typhoons in shallow water that are approaching landfall. Nepartak is a potential example of shear induced mixing causing rapid cooling in deepwater, so it is worth the effort to investigate.

Reply: We thank the reviewer for reminding us the nice work done by Glenn et al. (2016). We found that this paper is of particular importance to strengthening the interpretation of the manuscript. In our results and discussion, We referred that the dynamics for hurricanes (or typhoons) induced rapid upper ocean cooling in shallow shelf regions has been detailed in the paper of Glenn et al. (2016), and applied their finding to the analysis of our observations in deep water for Nepartak somewhat as an extension of Glenn et al.’s work. Importantly, results from a numerical study using the one-dimensional Price-Weller-Pinkel model (PWP model) has been incorporated in the revised manuscript to support the dynamics we inferred for the observed rapid temperature during the forced period of Nepartak. The simulations (i.e. temperature and velocity in the upper ocean), with some of the observations and wind stress and air-sea heat fluxes calculated from the observations as initial and boundary conditions, agree well with the corresponding observations, justifying the robustness of the PWP model in facilitating our dynamical interpretation of the data. A detailed discussion of the numerical simulation has now been added in Discussion. Indeed, we also adopted a one-dimensional model with Mellor and Yamada’s level 2.5 turbulence closure scheme (Mellor and Yamada, 1982) (M-Y model) to examine the dynamic processes for the fast upper ocean response to a super typhoon (see the model results in the figure below). To find the best parameterization in a model for the simulation of super typhoons, we are evaluating the performance of these models which consist of different turbulence determining schemes and drag coefficients for the calculation of wind stress in the model. We will definitely seek another opportunity to publish the detailed results from the PWP and M-Y models.

Numerical results from the one-dimensional M-Y model. The panels from top to bottom show time series of the surface wind stress, temperature in the upper 100 m with the temperature anomaly indicated by dashed lines, current velocity of u and v.

- 3) Line 25: Where is the discussion of the “temperature deviation developed downward to 500 m in 5 hours”? I see figure 4 plots extend to 500 m depth. What in these plots is this statement referring to? I see little difference before, during or after the storm at 500 m at short times scales, and some differences at NTU2 at 500 m a few days after the storm.

Reply: We improved the color code for Figs. 3d and 3h (see the new Figure 3) to enhance inertial motion relevant significant temperature anomaly, which is now defined as $|\delta T| > 0.2^\circ\text{C}$. The associated description has been rephrased. Inertial motion induced significant temperature deviation, defined as $|\delta T| > 0.2^\circ\text{C}$, began at approximately $-1.25 \times \text{IP}$ (when the circle of the Beaufort scale 7 wind was still very far away from NTU1 and NTU2), and δT developed quickly to the water column between thermocline and ~ 200 m as the eye center of typhoon passed nearby the buoy, particularly at NTU2. The sentence in question “temperature deviation developed downward to 500 m in 5 hours” has been deleted in this revision.

- 4) Line 26-28: Modification of the observed near-inertial period by the relative vorticity of the background flow is well established. What insights does this provide on the upper ocean response, especially during the direct forcing stage when the upper ocean can feedback on Nepartak’s intensity?

Reply: We agree with the reviewer’s comment. The discussion on the modification of near

inertial period has been reduced in this revision. Since Nepartak was a fast-moving typhoon with translation speed of about 14 km/h, the feedback of inertial oscillation-induced temperature decrease in the upper ocean on the intensity of Nepartak was limited. In the revised manuscript, we just report the existence of the near-inertial motion prior to the arrival of typhoon wind at the buoys and hypothetically attribute the cause of the early near-inertial motion observed by the two buoys to barotropic response of ocean to the low air pressure center of Nepartak. The redshift in the frequency of near-inertial motion at NTU1 is distinct, which is not successfully explained by the influence of ambient relative vorticity. The cause for this pronounced redshift is worthy of a further study. The surface temperature variation caused by the near-inertial motion before the typhoon does likely not influence the strength of Nepartak.

- 5) Lines 56-68: This paragraph describe the findings from similar deepwater met-ocean moorings in Hurricane Fabien, Typhoon Choi-Wan and Typhoon Kalmaegi. Perhaps a table noting the maximum wind speed observed by the mooring in each case and the closest approach of the eye center would help put the Nepartak observations in context and help support statements like “unprecedented data set” on line 81 and “the only in situ high resolution observation of a super typhoon, with its center passing near two data buoys” on lines 87-88.

Reply: With the reviewer’s suggestion, we added a table comparing maximum storm wind speed (W_{\max}), the closest distance between the storm center and the buoy (L_{\min}), and parameters observed in the upper ocean in deep water. The new table is also pasted below for the reviewer's reference.

Table 1. Comparison of deep water buoys observed tropical cyclones. “C” means category based on the Saffir-Simpson scale, Depth is the water depth (in meters) at where the buoy was anchored, L_{\min} is the closest distance (in km) from the buoy to the typhoon/hurricane eye center, and W_{\max} is buoy observed maximum wind speed (in m s^{-1}).

Tropical cyclones			Mooring characteristics				
Year	Name	C	Location of buoy	Depth	Parameters	L_{\min}	W_{\max}
1975	Eloise	2	Gulf of Mexico	2,541	T at 2, 50, 200, and 500 m; (u, v) at 50 m.	16	35
1988	Nelson	2	southeast of Japan	4,900	T at 8 depths in the upper 100 m.	50	43.5
1995	Felix	1	Bermuda	4,567	T at 7 depths (upper 150 m); (u, v) at 25, 45, 71, and 106 m.	90	38
2003	Fabian	3	Bermuda	4,567	T at 16 depths (upper 1500 m); (u, v) profiles (upper 200 m).	102	54
2005	Nate	1	Bermuda	4,567	Same as Fabian in 2003	123	39
2009	Choi-Wan	1	Kuroshio Extension	6,000	T at 20 depths (upper 525 m)	40	30
2010	Fanapi	3	western North Pacific	5,450	T at 14 depths (upper 147 m)	68	18
2010	Megi	5	western North Pacific	5,500	T at 10 depths (upper 148 m)	123	23.5
2014	Kalmaegi	1	South China Sea	3,990	T at 15 depths (upper 400 m); (u, v) profiles (upper 245 m).	32	23

2016	Nepartak	5	western North Pacific	5,490	T at 12 depths (upper 500 m); (u, v) at 25 and 75 m.	19	46.5
2016	Nepartak	5	western North Pacific	4,870	T at 12 depths (upper 400 m); (u, v) at 25 and 75 m.	6	44.0

6) Line 61. This manuscript lists Typhoon Choi-Wan as Category 5. Bond et al., 2011 (Ref #5) lists it as Category 1 on their page 3. Figure 4 in Bond et al. (2011) also shows the most significant cooling occurring during the upwelling of the thermocline just after the eye center passage. The Nepartak data is different, with significant cooling before eye center passage while the thermocline is deepening. These details in the Nepartak observations that make them different from observations in other extreme storms are difficult to explore in the existing figures.

Reply: We corrected the typo for the Saffir-Simpson scale of Choi-Wan. We thank the reviewer for the in-depth evaluation of the observations between Cat. 1 Choi-Won and Cat. 5 Nepartak, and the corresponding figures showing the observational data have been improved as possible as we can to respond the reviewer's concern.

7) Line 83: This paper examines the measured "ocean mixed layer" and computed "air-sea heat fluxes". Air-sea fluxes are not measured, but are computed based on the TOGA COARE bulk algorithms. There is increased uncertainty in these algorithms at very low and very high wind speeds.

Reply: We have modified our interpretation of the data and the variables derived from the data. The sentence has been revised as "A data set of atmospheric and oceanic variables collected by our two buoys during the passage of Super Typhoon Nepartak in July 2016 provides a unique opportunity to gain an in-depth understanding of the evolution of temperature and current responses to extremely strong typhoon winds in the upper ocean and to validate theories and numerical forecasts of super typhoons." (See line 84-88.)

8) Line 88-90: How does the video contribute to the new insights on the upper ocean response? Maybe stills during specific phases of the storm can provide these insights? Some discussion of the video and how it is relevant to the interpretation of the upper ocean response is required if the video is to be included.

Reply: The time-lapse camera mounted on the tower of the buoy actually provided an "on-site" look of the sea surface condition. This was also a unique chance for us to look at the air-sea interface during the influence of Category 5 typhoon winds. In the revised text, we highlighted that the time-lapse camera visualized the sea surface while Nepartak was approaching the buoy, providing an "on-site" look of the sea surface condition. The air-sea interface became indistinguishable during the extreme strong wind, conceivably hampering the estimation of air-sea fluxes. The blurred air-sea interface is certainly one of sources for the uncertainty in the TOGA COARE algorithms at extreme high wind speed. If the reviewer still feels the video cannot be regarded as part of the unprecedented data set, we will exclude the video record from Supplementary.

9) Lines 96 & 98. Here it says closest approach of Nepartak to the buoy NTU1 was 15 km and

buoy NTU2 was 5.7 km. Abstract says 19.2 km and 5.9 km. Zhang et al. (2016) (Ref #8 & #28) also discuss the track uncertainty from different sources. Data section does not say which source for the track was used, and does not include any measure of the uncertainty in the track at the time of closest approach to the moorings.

Reply: The closest distances from the eye center of Nepartak to the two buoys in the manuscript have been corrected. We further clarified (in Data) that the typhoon track was provided by the Central Weather Bureau, which was positioned from satellite imagery of Japanese satellite Himawari-8 with $0.1 \times 0.1^\circ$ resolution (~ 11 km at 20°N) every 3 hours during “normal weather” and every 1 hour when the typhoon is over the sea and land. The maximum error in the track estimation is thus ~ 15 km. How the location of the track and the eye center of the typhoon is determined has been addressed in Data section. (See line 455-458.)

10) Line 102. That the cold wake is 2.5C cooler than the surrounding water is an interesting observation. But the main purpose of this paper is the upper ocean response, especially during the direct wind forcing section. I am not sure which SST images are cloud free, but a before, after and difference SST map is almost standard for this. Also, the source of the SST data is not described in the data section.

Reply: The satellite sea surface temperature is obtained from the microwave Optimally Interpolated (OI) SST, which is available at <http://data.remss.com/SST/daily/>. The image of satellite SST difference obtained from the subtraction of the post-storm SST from the pre-storm SST in Fig. 1 (below) has been improved to clarify the temperature decrease and spatial coverage of the cold wake.

Figure 1. (a) Satellite SST difference between pre-storm SST (5 July) and post-storm SST (8 July) with the typhoon track indicated by the gray line. Dashed circles and white contours in (a) indicate the radius of the Beaufort scale 10 wind and sea level anomalies of -0.1 and -0.2 m. Buoy NTU1 observed (b) air pressure (black line), wind speed (blue line), (c) temperature in the upper 300 m, and (d) the meridional current velocity V_{curr} at 75 m (black line) and meridional wind velocity V_{wind} (red line). The two red vertical lines in (b)–(d) mark the beginning (left) and end (right) of the forced period. The black curve in (c) is the 27.2°C isotherm. (e) Observed (red line), simulated (blue line), and pre-storm (black dashed line) temperature profiles at times 1–6 as indicated by the vertical black dashed lines in (c) and (d). Green lines in (e) mark the mixed layer depth. The green vertical line in (b)–(d) marks the time of minimum sea level pressure.

- 11) Line 104. East of 124E, stating that the cold water on the left-hand side of the track is in the same location as the cold eddy is defensible. Saying it is “due to” the pre-existing cold eddy modulating the shape of the cold wake, a cold wake that is related to the inertial response, is unsupported speculation that could be investigated with a model.

Reply: There were indeed several observations and associated studies suggesting that a pre-existing cold eddy could modulate the typhoon-induced cold wake as that presented in the SST anomaly after Nepartak. An example has actually demonstrated and delineated in Lu et al. (2016). We added this paper as a reference among others to the discussion of the satellite SST anomaly after the typhoon in the revised manuscript. (See line 116-119.)

Reference:

Lu, Z., Wang, G., & Shang, X. (2016). Response of a preexisting cyclonic ocean eddy to a typhoon. *Journal of Physical Oceanography*, 46(8), 2403–2410.

- 12) Lines 105. A focus on the upper ocean temperature and velocity response during the forced period remains an excellent focus for any revisions of this paper. Relating this to the storm intensity would be useful. It may be as simple as noting if the intensity is remaining steading or how it is changing during this time period.

Reply: We significantly strengthened the interpretation and discussion on the rapid upper ocean temperature and velocity response to the direct wind impact period of Nepartak as suggested. We noted that the strength of Nepartak remained Category 5 between 06:00 6 July UTC and 12:00 8 July UTC before it touched the southeast coast of China, suggesting no significant intensity change during the observations of the two buoys. See the first paragraph of Results (line 104-126).

- 13) Lines 109-111. The definition for the forced period used here (start of temperature response and end of shear instability development) is not based on the forcing, but on the response. This should be reconsidered. The strong forcing can start before the temperature responds, and the change in vertical shear is also influenced by the deepening of the pycnocline. The reduction in the observed shear used to define Stage 2 is at least partially related to the pycnocline migrating below the bottom current meter so that both current meters are eventually in the upper layer with little shear observed. There still may be significant shear between the deepening surface layer and the deeper layer below, something that a model could reveal.

Reply: We take the reviewer's suggestion and have now re-defined the forced period in accordance with the variation of typhoon wind and the arrival of the eye of Nepartak. We have also declared in Discussion that as the development of the mixed layer deeper than 80 m quickly the two current meters fixed at 25 and 75 m were apparently not adequate to resolve the critical velocity shear above and below the thermocline. The PWP model we adopted as suggested did provide a plausible support to our inference on the development of velocity shear which causes fast deepening of mixed layer during the forced period of Nepartak. We believe that the associated description has been significantly improved.

14) Lines 115-119. The details of the thermal structure described here would be greatly aided by some profile plots. Sample profiles at critical times in the storm could be shown to aid the discussion throughout.

To respond the reviewer's suggestion, we modified Figs. 1 and 2 with adding temperature profiles at 6 time points covering the direct influence period of typhoon wind, including before and after the forced period. The evolution of temperature change in the upper layer is clearly seen from these profiles. The revised Fig. 2 is pasted below for the reviewer's evaluation.

Figure 2. Same as Fig. 1 but for NTU2, with the exclusion of satellite SST derived variables and the addition of V_{curr} at 25 m (thick black line) and 75 m (thin black line) in (c) and the velocity shear square \bar{S}^2 (red line), four times of buoyancy frequency square $4\bar{N}^2$ (blue line), and $\bar{S}^2/4\bar{N}^2$ (green line) in (e). The two white lines in (b) indicate the depths of the two current meters measured by the pressure sensors of the current meter.

15) Line 122 & 141. Stage 1 & 2 definitions during the forced period are based on the responses noted above, which have some inherent flaws, and are only definable for one of the moorings

due to current meter failures. Glenn et al. (2016) also divide the forced period into two stages, ahead-of-eye-center and after-eye-center based on the air pressure record to define the eye center and the wind speed to define the start and end of the forced period. Adopting these definitions will enable the upper ocean response to be examined for both moorings during a forced period that can still be divided into two relevant stages. For example, there is a significant change in the TOGA COARE computed heat fluxes immediately after eye passage based on the air pressure record in both moorings. Time series of SST and air temperature to accompany the existing time series of air pressure and wind speed could provide further insights as to why.

With the reviewer's suggestion and the definition in Glenn et al. (2016) in mind, we redefined the forced period and associated Stage 1 and 2 in the second paragraph in Results (line 125-130). The forced period is now determined from the beginning as the wind speed increases rapidly to 17 m s^{-1} to the time as the wind speed decreases to 17 m s^{-1} (marked by red dashed line in Figs. 1b and 2a). Our observations find that the evolution of upper ocean responses before and after the arrival of the typhoon center at the buoy differs significantly as that reported in Glenn et al (2016), and therefore the forced period is split to Stage 1 and 2 by the time that the minimum air pressure is recorded by the buoy.

- 16) Lines 143-146. The way to demonstrate that “turbulent mixing induced by shear instability, rather than air-sea heat flux, dominates the seawater cooling” is to use a model and look at the terms in the equations for the heat balance.

Reply: As our response to reviewer's comment 2), results from the one-dimensional PWP model simulation greatly help support our dynamical speculation for the development of turbulence mixing from the critical shear instability. The heat loss from the sea surface to the atmosphere during the forced stage of Nepartak is proven by the numerical simulations with or without air-sea heat exchange to be very minor in the rapid upper ocean temperature decrease. The numerical simulations suggest that the temperature drop induced by the air-sea heat flux is one order of magnitude smaller than the extreme strong wind induced turbulence mixing does.

- 17) Line 145. N^2 is introduced here, and the process to calculate it is described in lines 416-418. But there is only the Figure 1 plot of the temperature data, with no similar plot or discussion of the salinity or density field. It is understood that *Nature* figures are supposed to be compact, but even a short description of the salinity or density time series, or salinity and density plots similar to 1b and 1g in the supplemental materials, would help.

Reply: As the reviewer suggested, we added a brief description of the density variation in the paragraph (line 191-192). The variation of density essentially follows the variation of temperature during the forced period of Nepartak.

- 18) Lines 153-155. As noted above, there are multiple reasons why the $4N^2-S^2$ is small during Stage 2. Early on in the storm, the 25 m and 75 m observation depths are on different sides on the pycnocline. As the pycnocline deepens during the storm, the observation depths are increasingly both in the upper layer. This is a fatal flaw when trying to use just the observations to describe the processes. Using the observations to validate a model that is then used for interpretation of processes is a more likely to be successful.

Reply: Please see our response to reviewer's comment 2) and 16).

19) Lines 158-178. Interpretation of Figure 2 supports the above alternate interpretation of the current meter record. During Stage 1, the currents are in difference directions, and the temperatures differ by about 4C. During Stage 2, currents are in the same direction and temperature difference quickly goes to zero, implying the 2 current meters are in the same layer.

Reply: See our response to the comment 13). We have added a note (see line 374-380) to indicate that, as the development of the mixed layer from ~30 m to deeper than 80 m quickly, the two current meters fixed at 25 and 75 m cannot resolve the critical velocity shear above and below the thermocline. In this light, we take the reviewer's recommendation to use the PWP model for investigating the evolution of the shear instability, particularly at depths deeper than 75 m, during the forced period. In the model simulation for NTU2, the shear instability is still noticeable at depths deeper than 75 m during Stage 2 (see new Fig. 5e). The model result compensates the limitation of our current meter observations at fixed depths and lends support to our speculation for the development of shear instability and turbulent mixing during the direct strong wind influence period of Nepartak.

20) Line 183. What does the depth variation of the current meter (gray line) mean? Is this a measure of the waves?

Reply: This version of the buoy did not have a wave gauge. The gray line means the depth of current meter in the original figure and the depth variation is caused by the heaving of the current meter during the forced period. Since large typhoon waves can conceivably push the buoy toward the wave direction, the mooring line was significantly tilted and the current meter was therefore heaved when the typhoon center was near the buoy and restored to its original depth after the passage of typhoon center. We revised legend of Fig. 2, and the two white lines in Fig. 2b indicate the depths of the two current meters measured by the pressure sensors of the current meter.

21) Line 217. Caption says "observed" heat flux. It is computed based on TOGA COARE.

Reply: We have corrected the expression for how the air-sea heat flux is obtained throughout the revise manuscript.

22) Lines 221-226. Is the comparison of the computed air-sea fluxes from the buoy with the time series derived from the OAFflux maps necessary? The OAFflux maps are not expected to resolve the storm, and this shows they do not. How does this result provide new insights into the upper ocean response? It does not seem necessary. It is certainly not a conclusion that one would publish in *Nature*.

Reply: We agree with the reviewer's comment. Therefore, we have excluded the comparison of buoy data derived air-sea heat fluxes and those provided by OAFflux during extreme strong winds of Nepartak in this revision.

23) Lines 230-235 explain some uncertainties in the TOGA COARE algorithms, that they may

not apply at high wind speeds, which is why they are not direct observations of the heat flux. Some discussion of this uncertainty will help show that the general relative magnitudes and trends in the fluxes are still valid.

Reply: We added a comment on the air-sea heat fluxes derived from the buoy data and provide the video taken by the time-lapse camera to suggest that the order of magnitudes of the estimated heat fluxes are valid and the blurred air-sea interface during the strong wind of Nepartak causes known uncertainties to the estimate of air-sea heat fluxes using existing algorithms such as the TOGA COARE 3.0. The comment is added in the first paragraph of “Buoy data derived air-sea heat flux and near-inertial motion” as:

The magnitudes of heat fluxes derived from the buoy data, although not surprising, are helpful to validating the numerical simulation for Nepartak. It is also seen from the on-site images (Supplementary Video) that the interface of air and sea becomes foggy and blurred, forming a two-phase transitional layer, which creates difficulties in determining, e.g., the air temperature, wind speed, and specific humidity at 10 m above the “sea surface”. Since a portion of the wind momentum is used to speed up the sea spray above the sea surface, the drag coefficient can be increased slowly^{17,18}. Therefore, the uncertainty in the calculation of Q_{sen} and Q_{lat} is increased because the change in the drag coefficient during extreme strong winds is still not well-understood under disrupted air-sea interface.

24) Lines 236-299. Inertial response of the ocean to typhoons is well documented as noted by the authors in references 3, 21, 22, 23, 24, and 25. What new insights does this provide, and how does it contribute to the understanding of heat flux and velocity shear impacts on the ocean response?

Reply: What we originally like to point out is that there is rarely observation for the inertial motion which is on or very close to the typhoon track. The two buoys observed inertial motions in temperature profiles before the arrival of Nepartak’s eye, which is presumably induced by the remote effect of barotropic response of ocean to Nepartak (Shay and Elsberry, 1987; van Haren et al., 2018). The dynamics for this early inertial response remains to be examined. (See line 319-323.)

References:

Shay, L. K., & Elsberry, R. L. (1987). Near-inertial ocean current response to hurricane Frederic. *J. Phys. Oceanogr.*, 17, 1249-1269.

van Haren et al. (2018), Typhoon impact on deep-ocean bottom. *Journal of Geophysical Research Oceans* (submitted)

25) Line 276 (and others): The discussion refers to time in reference to inertial periods, but the time axis in the figures is labeled in days during the month of July. This makes it difficult to use the figure to understand and verify the discussion points. Can the inertial period time line be added to these figures to better follow the discussion?

Reply: The associated time axis in local inertial period was originally plotted on top of Fig. 4 (now Fig. 3 in the revised manuscript).

Figure 3. Buoy observed air pressure (black line) and wind speed (blue line) at (a) NTU1 and (e) NTU2 during 4–14 July 2016. The top axis of (a) and (e) indicates time in inertial periods. The wind speed 13.9 m s^{-1} (equivalent to 50 km h^{-1} or a Beaufort scale 7) is represented by a blue horizontal line in (a) and (e). Buoy data-derived net air-sea heat flux (Q_{net}), net radiation ($Q_{sw}+Q_{lw}$), latent heat flux (Q_{lat}), and sensible heat flux (Q_{sen}) at (b) NTU1 and (f) NTU2. Observed velocity-derived 75 m depth inertial-band (periods from 28.07 to 49.3 hours) velocity sticks at (c) NTU1 and (g) NTU2. Inertial-band temperature anomalies in the upper 500 m (color shading) and the mixed layer depth (blue lines) at (d) NTU1 and (h) NTU2. The two pink vertical dashed lines indicate the time at a minimum air pressure (left) and after the 0.5 local inertial period (right).

26) Lines 306-308. This is an important conclusion, that the vigorous mixing leads to the rapid decrease in sst during the direct forcing stage while the storm is still present, and that negative feedback can weaken the storm. Focusing on this result, and comparing it to previous observations and modeling studies has the potential for new insights. For example, for tropical storms in deepwater, the Price models indicate that the rapid cooling during the direct forcing period is expected to be approximately evenly split between ahead-of-eye-center cooling and after-eye-center cooling. Is this what is observed in Nepartak? It is difficult to tell from Figure 1 alone.

Reply: The observed rapid temperature drop during the forced period of Nepartak is not evenly split between ahead-of-eye-center cooling and after-eye-center cooling. Our observations show that the evolution of upper ocean cooling and velocity variation between ahead-of-eye-center and after-eye-center differs significantly. Therefore the forced period is split into Stages 1 and 2 by the time that the minimum air pressure is recorded by the buoy. The forced period at

NTU2 (14.3 hours) is longer than that at NTU1 (9.25 hours) because NTU2 experienced almost the full diameter of Nepartak's storm circle, whereas NTU1 was on the secant line of the storm circle. Figure 1 has been significantly improved and we hope the difference between the two-buoy observations and the Price models is clearly seen. (See line 127-134.)

27) Lines 315-317. A model result would demonstrate this.

Reply: Please see our response to the reviewer's comment 2) and 16).

28) Lines 318-320. This process of the surface layer accelerating in response to the wind resulting in vertical shear between the surface and deep bottom layer that then produces enhanced mixing is a known process in the Price models.

Reply: We agree with the reviewer. This statement has been deleted in this revision. Instead, we detailed the evolving process of the upper ocean temperature cooling and underlying dynamics on the basis of the unprecedented observations of the two buoys for the super typhoon Nepartak. The data set is helpful to validating existing theories and process-oriented numerical studies for the rapid upper ocean thermal and velocity response to extreme strong typhoon/hurricane winds.

29) Lines 324-325: Not sure where the temperature variations developing downward to 500m within 5 hours is discussed.

Reply: The description was originally based on our evaluation on the time series of temperature profiles (not shown in this paper) at the two mooring locations. After a thoughtful discussion between authors, we decided to remove this observation and to concentrate our interpretation and discussion on the fast temperature variation in the upper 300 m. Even so, we can still observe the temperature decrease below the mixed layer particularly after the eye center of Nepartak, presumably due to the Ekman suction (Zhang et al., 2016). We added a modest discussion for the deepwater column cooling in line 372-382.

30) Lines 332-334: I would expect that the maximum inertial band temperature variation below the deepening thermocline is the oscillation of the thermocline depth as has been observed in the inertial tail of other typhoons. But the blue line in Figure 4c and 4f for the mixed layer depth does not oscillate. Is the blue lined smoothed?

Reply: The reviewer's suggestion for the maximum inertial band temperature anomaly is correct. The blue lines in new Figure 3 are smoothed by a 48-hour low pass filter, and therefore the depth variation of the maximum inertial band temperature anomaly is clearly seen to follow the variation of mixed layer depth. The relationship of two depth variables is the focus of our follow-on study of this data set.

31) Lines 343-355: This uses geostrophic currents from satellite derived sea surface height to calculate the background relative vorticity to see if this is responsible for the shift in the observed near-inertial frequency. Good agreement is achieved with NTU2, which, based on supplemental figure S1, is located between the Kuroshio and the anticyclonic mesoscale eddy. Agreement is not as good with NTU1, which is located between two mesoscale eddies of opposite signs. While Figure S1 is not sufficient to tell the exact locations of the buoys

relative to these ocean features, it seems that errors in the location of the two eddies can have a large impact on the background vorticity estimate at NTU1, even changing the sign of the relative vorticity. This could be investigated as potential explanation of the discrepancy between the observed near inertial frequency at NTU1 and the frequency estimated here. A map of the geostrophic current fields or sea surface height contours with the buoy locations properly marked would be a first step, and is easily included in the supplemental materials.

Reply: To clarify the reviewer’s concern, we added a map showing the relative vorticity calculated from the satellite SSH-derived absolute geostrophic current and the buoy locations to Supplementary Fig. S1 (Fig. S1d and below). The hypothesis of effective f ($f_{\text{eff}} = f_0 + \zeta/2$) explains the redshift in the frequency of the near-inertial oscillation derived from the observations better for NTU2 than for NTU1. The frequency of near-inertial motion derived from the data at NTU1 may be modified more by the nonlinear effect raised by the consistence of the typhoon translation speed ($\sim 3.89 \text{ m s}^{-1}$) and the phase speed of the first baroclinic mode and the vertical variation of the ambient relative vorticity (see the discussion at line 298-314).

Supplementary Figure S1d. Relative vorticity (color shading) of the geostrophic flow field in Fig. S1b.

Minor typos:

Line 35, “forecast” should be “forecasts”. Line 37, what are “false announcements of the typhoon day”? The forecast landfall day? Line 42, “inertia” should be “inertial”. Line 133, air pressure is light blue and wind speed is orange. Line 138, what are the white lines in Figure 1e? Line 243. Units for inertial frequencies are usually 1/seconds. Line 250. Notation of “ $t = -1.25 \cdot IP$ ” is confusing. I interpreted this as $t = -1.25 \cdot IP$ Line 509. “white” line.

(1) These typos and sentences in question have been all corrected as suggested.

(2) We have clarified “false announcements of the typhoon day” as follows. “Using in situ observations to advance our knowledge of air-sea exchanges during extremely strong winds and, in turn, improve the accuracy of the numerical typhoon forecasts is of particular importance to providing timely warnings to the public for disaster mitigation and for reducing

economic loss that results from false announcements. These false announcements are mostly attributed to forecasting errors in a storm's landfall time/location and typhoon wind strength." (see line 25-30)

(3) The unit of inertial frequency has been changed to s^{-1} . To further clarify, "t = -1.25-IP", e.g., has been changed as "t = -1.25 IP".

Responses (in blue) to reviewer #2's comments.

Reviewer #2 (Remarks to the Author):

This paper describes some very useful observations made a new kind of deep sea data buoy that is able to survive encounters with very intense storms. The data are therefore quite valuable, and I do believe that this paper is worthy of publication in a very high quality international journal. I am guessing that the editors will ask that the paper be made more concise, and I encourage that as well; omit everything that does not directly aid the reader.

With the reviewer's constructive suggestion, we have conducted a thorough revision for the manuscript and deleted discussions, such as the comparison of observational data-derived surface heat flux and that from OAFlux, that are less novel in our original manuscript. The unique dataset details the temporal and spatial evolution of rapid upper ocean temperature and velocity response to the super typhoon Nepartak during its forced period, and is helpful to validating associated theories and numerical studies/forecasts for the air-sea interaction during extreme strong typhoon/hurricane winds. We have taken each of the reviewers' comments to improve all the figures and the interpretation of each figure. Importantly, the associated dynamics we inferred from the data set is further examined using the one-dimensional Price-Weller-Pinkel (PWP) model with the observations and variables derived from the data as the initial and boundary conditions. The numerical simulations provide promising support to our inference of the dominant role of the typhoon-enhanced velocity shear instability in the rapid upper ocean cooling and concurrent deepening of the mixed layer.

Some details regards the discussion.

1) Suggest that you check to see how big the temperature change due to the surface heat flux should be (evaluate a 1-d heat budget). No doubt that the surface heat flux contributes to SST cooling, but I expect that it will not be a very large fraction of the total cooling even during the first stage.

Reply: Our discussion on the cause of rapid temperature drop in the forced period of Super Typhoon Nepartak is actually a little vague in the previous manuscript. In this revision, we adopted the one-dimensional PWP model simulation to supplement our explanation for the dynamics of the temperature decrease. Not surprisingly, the heat loss from the sea surface to the atmosphere during the forced stage is proven to be very minor in the rapid upper ocean temperature decrease. The simulations with or without air-sea heat fluxes suggest that the temperature drop due to sea surface heat loss to the atmosphere is one order of magnitude smaller than that caused by the extreme strong wind induced turbulence mixing (approximately 0.2°C vs. 2°C).

2) Estimating the effect of shear on stability is worthwhile, but is somewhat limited since only two depths are available. Very likely the stability was lower at other depths not sampled. Nothing you can do about that other than be aware of it and take care not to over-interpret these data. Specifically, higher than critical stability between 25 and 75 m does not foreclose the possibility that stability was lower elsewhere and contributed to vertical mixing during the first stage.

Reply: We agree with the reviewer’s comment and thus have added a note (see line 374-380) to indicate that, as the development of the mixed layer from ~30 m to deeper than 80 m occurs quickly, the two current meters fixed at 25 and 75 m cannot resolve the critical velocity shear above and below the thermocline. The PWP model we used to simulate the upper ocean temperature and velocity response provides a reasonable evolving process of the shear instability, particularly at depths deeper than 75 m, during the forced period. In the model simulation for NTU2, the shear instability is still noticeable at depths deeper than 75 m during Stage 2 (see new Fig. 5e in the revised manuscript or below). The model result compensates the limitation of our current meter observations at fixed depths and lends support to our inference on the development of shear instability and turbulent mixing during the direct strong wind influence period. We are also working on the one-dimensional dynamics of the fast upper ocean cooling during the direct forcing period of Nepartak using the other one-dimension model with the Mellor and Yamada (1982) level 2.5 turbulence closure scheme as the parameterization of mixing in the model (M-Y model). The M-Y model simulated upper ocean temperature variation under the influence of extreme strong wind is demonstrated in the figure below. To find the best parameterization in a model for the simulation of super typhoons, we are evaluating the performance of these models with different turbulence determining schemes and drag coefficients for the calculation of wind stress in the model, and will seek another opportunity for the publication of our evaluation.

Fig. 5. Vertical profiles of (a) PWP model-produced temperatures, model data-derived (c) shear square S^2 , (d) four times of buoyancy frequency square $4N^2$, and (e) $S^2/4N^2$ at NTU2. (b) Time series of the upper 25 m-averaged temperatures obtained from the observations (T_{obs}), from the PWP model-produced temperature (T_{pwp1}), and from similar model settings but with a zero surface heat flux-produced temperature (T_{pwp2}). The two horizontal white lines in (e) mark the depths of the two current meters.

Numerical results from the one-dimensional M-Y model. The panels from top to bottom show time series of the surface wind stress, temperature in the upper 100 m with the temperature anomaly indicated by dashed lines, current velocity of u and v.

3) The discussion of the deep temperature response (lines 323+) is a bit muddled. It almost sounds as if it is an extension of the cooling/mixing processes that cause SST cooling. However, the deep response is obviously reversible (it oscillates) and so is not due mainly to a mixing process. Rather, it is almost certainly associated with inertial pumping that you made reference to earlier in the paper.

Reply: We clarified the possible cause for the temperature variation in the deep layer. The temperature decrease below the mixed layer particularly after the arrival of the eye center of Nepartak is most likely due to the Ekman suction (the Ekman pumping) (Zhang et al., 2016). A modest discussion for the deepwater column cooling is inserted in line 381-391.

4) Not sure I see the value of the horizontal shear effects on the inertial oscillation frequency. Unless I misunderstood, the estimated effect is much less than was actually observed?

Reply: We clarified the discussion for the possible cause of the redshift in the frequency of near-inertial oscillation derived from the observed velocity (see line 305-314). The relative vorticity (from the horizontal shear) induced inertial oscillation frequency shift is consistent with the redshift derived from the observational data at NTU2 (-4.8% vs. -3.29% of the local f_0 at NTU2). The estimated relative vorticity effect in the shift of inertial frequency is much significant than the actual frequency shift derived from the observed velocity at NTU1 (-13.3% vs. -0.53% of the local f_0 at NTU1). The redshift of the observed inertial oscillation frequency at NTU1 definitely warrants a further study.

Reviewer #1 (Remarks to the Author):

I congratulate the authors on a job well done. All of my comments have been addressed as indicated in the response to reviewers. The improved figures and the addition of the PWP model results have greatly improved the quality of the paper. The unique observations combined with the modeling results now clearly illustrate an important deep water case study for the role of shear induced mixing resulting in rapid upper ocean cooling during the direct forcing portion of a high energy typhoon. Especially noteworthy is the difference between the rapidly deepening high shear layer in Stage 1 and the relatively constant depth of the shear layer in Stage 2. The authors themselves state that this has already inspired their own future modeling studies, and it will inspire others. The authors have also provided an important reason for including the video in the supplemental materials. I trust that a copy editor will clean up some English on lines 88 (needs plurals) and 412-413 (wind influence ineffective layer). This paper is good to go.

Reviewer #2 (Remarks to the Author):

This is a very nicely written and presented paper, and I believe worthy of publication in Nature Comm.

The only suggestion that I will make on this revised ms is to remove the last sentence of the abstract. This is not a main theme of the paper and not something that is treated in any depth in the analysis.

James F Price.